# Efficacy of Popular Diets Applied by Endurance Athletes on Sports Performance: Beneficial or Detrimental? A Narrative Review

**DOI:** 10.3390/nu13020491

**Published:** 2021-02-02

**Authors:** Aslı Devrim-Lanpir, Lee Hill, Beat Knechtle

**Affiliations:** 1Department of Nutrition and Dietetics, Faculty of Health Sciences, Istanbul Medeniyet University, 34862 Istanbul, Turkey; asli.devrim@medeniyet.edu.tr; 2Division of Gastroenterology & Nutrition, Department of Pediatrics, McMaster University, Hamilton, ON L8N 3Z5, Canada; hilll14@mcmaster.ca; 3Medbase St. Gallen, am Vadianplatz, 9001 St. Gallen, Switzerland; 4Institute of Primary Care, University of Zurich, 8091 Zurich, Switzerland

**Keywords:** diet, fat, carbohydrate, protein

## Abstract

Endurance athletes need a regular and well-detailed nutrition program in order to fill their energy stores before training/racing, to provide nutritional support that will allow them to endure the harsh conditions during training/race, and to provide effective recovery after training/racing. Since exercise-related gastrointestinal symptoms can significantly affect performance, they also need to develop strategies to address these issues. All these factors force endurance athletes to constantly seek a better nutritional strategy. Therefore, several new dietary approaches have gained interest among endurance athletes in recent decades. This review provides a current perspective to five popular diet approaches: (a) vegetarian diets, (b) high-fat diets, (c) intermittent fasting diets, (d) gluten-free diet, and (e) low fermentable oligosaccharides, disaccharides, monosaccharides and polyols (FODMAP) diets. We reviewed scientific studies published from 1983 to January 2021 investigating the impact of these popular diets on the endurance performance and health aspects of endurance athletes. We also discuss all the beneficial and harmful aspects of these diets, and offer key suggestions for endurance athletes to consider when following these diets.

## 1. Introduction

Endurance performance, especially prolonged training, requires greater metabolic and nutritional demands from athletes [1]. As endurance athletes face harsh conditions during training periods, they seek alternative dietary strategies to improve endurance performance and metabolic health [2]. It is of paramount importance that a popular diet should be scientifically proven before being adopted in the athletic population [3]. Vegetarian diets [4], high-fat diets (HFD) [5], intermittent fasting (IF) diets [6], gluten-free diet (GFD) [7] and low fermentable oligosaccharides, disaccharides, monosaccharides and polyols (FODMAP) diets [8] are very popular among endurance athletes. In this review, we will discuss both the beneficial and harmful aspects of these diets on metabolic health and endurance performance.

## 2. Methods

We searched both the PubMed and Cochrane databases for the terms “diet*”, “track-and-field”, “runner*”, “marathoner*”, “cyclist”, “cycling”, “triathlete”, “endurance”, and “endurance athletes” in the title, abstract, and keywords to detect the most applied diets between 2015 and 2021 in endurance athletes. We obtained 217 results in PubMed and 80 trials in the Cochrane database. We defined the most recurrent diets in endurance athletes, including “High CHO availability”, “High-carbohydrate diet”, “Ketogenic diet”, “Low-CHO diet”, “Low-CHO, high-fat diet”, “Ketogenic low-carbohydrate, high-fat diet”, “Low-carbohydrate ketogenic diet”, “Low-carbohydrate, high fat, ketogenic diet”, “High-fat, low carbohydrate diet”, “Ketone ester supplementation”, “time-restrictive eating”, “Ketone supplementation”, “Intermittent fasting”, “fasting during Ramadan”, “Vegan diet”, “Lacto-Ovo vegetarian diet”, “Vegetarian diet”, “Low fermentable oligo-, di-, monosaccharide, and polyol diet”, and “Gluten-free diet”. Since we all know that high-carbohydrate diet is already well proven to enhance endurance performance [2], we targeted other diets for in-depth investigation by categorizing them as “vegan/vegetarian diets”, “high-fat diets”, “intermittent fasting”, “low-FODMAP diet, and “gluten-free diet”. We included studies on endurance athletes and popular diets, including vegetarian diets, high-fat diets, intermittent fasting, gluten-free diet, and low-FODMAP diet. Using PubMed, Cochrane Library, and Web of Science databases, we aimed to identify studies on races and endurance training. Two researchers (A.D.L and L.H.) independently reviewed the literature. In cases of conflict, a third investigator (B.K.) resolved the disagreement. We identified the studies published from 1983 to 2021. To define the studies on endurance athletes and diets to be included in the current narrative review, we searched MeSH terms ((“Diet, Ketogenic” (Majr); “Diet, High-Fat” (Majr); “Diet, Carbohydrate-Restricted” (Majr); “Ketone Bodies” (Majr); “Diet, Vegetarian” (Majr); “Diet, Vegan” (Majr); “Fasting” (Majr); “Diet, Gluten-Free” (Majr); “athletes” (Majr); “physical endurance” (Majr); “Diet Therapy” (Majr); “ Oligosaccharides” (Majr), “Disaccharides” (Majr)) and MeSH terms found below this term in the MeSH hierarchy recommended by PubMed and Cochrane Library. We also searched by adding the terms “FODMAP diet”, “low-FODMAP diet”, “FODMAP*”, “Fermentable oligosaccharides, disaccharides and polyols”, “Fermentable, poorly absorbed, short chain carbohydrates”, “Inulin”, “Xylitol”, “Mannitol”, “Maltitol”, “Isomalt”, “Fructose”, “Fructans”, “Galactooligosaccharides”, “fructooligosaccharides”, and “Polyols” to all databases, as no MeSH terms for the low-FODMAP diet were defined. We discussed the findings after determining the clinical and practical relevance of the studies by considering only human studies. We included studies available in English clearly describing the applied diet and investigating the effect of diet on endurance athletes as the primary goal. In addition, we included studies where diets were applied according to the dietary description. We excluded studies not explicitly addressing the impact of the diet on endurance performance or health-related parameters, that were not written in English, and were conducted on animals or in vitro. Based on our inclusion and exclusion criteria, we identified 57 research articles (Table 1). We organized the narrative review by considering both the beneficial and detrimental aspects of all five diets for endurance athletes.

## 3. Popular Diets Applied to Improve Sports Performance in Endurance Athletes

### 3.1. Vegetarian Diets

Worldwide, it is estimated that around four billion people follow vegetarian diets [9]. In addition to many books and documentaries on vegetarian diets along with various types of practice (Table 1) and many well-known athletes who have adopted vegan diets and improved their performance [10], vegan diets have become more acceptable and feasible in the athletic population [11]. Looking at the athletic population, using a survey-based study conducted with 422 marathon runners, approximately 10% (*n* = 39) of the athletes consumed vegetarian/vegan/pescatarian diets [12]. However, in the NURMI study, the authors used the prevalence of vegetarian diets in ultra-endurance runners, primarily living in Austria, Germany, and Switzerland [13]. The findings revealed that the ratio of vegetarian and vegan athletes was 18.4% and 37.1%, respectively.

#### The Impact of Vegetarian Diets on Sports Performance

Benefits of Vegetarian Diets

With the growing popularity of vegetarian diets in the athletic population, researchers have begun to investigate the role of these diets in sports performance and metabolic profile [71].

Studies on vegetarian diets have suggested that these diets may improve endurance performance by increasing exercise capacity and performance, modulating exercise-induced oxidative stress [72], inflammatory processes including anti-inflammatory and immunologic responses [4], and upper-respiratory tract infections (URTI) [73], and providing better cardiovascular function [59].

Studies measuring the aerobic capacity of vegetarian and omnivorous athletes reported controversial results [54,56,58,59]. Two studies showed that VO_2_max values were higher in vegetarian athletes compared to omnivore athletes [56,59], while a crossover study showed no difference between the groups [54]. Studies supported higher VO_2_max values in vegetarians designed as a case study and two cross-sectional studies [56,58,59], which are considered as the lowest level of the etiology hierarchy. A cross-sectional study in amateur runners reported that vegetarian female athletes had higher VO_2_max values than omnivorous female athletes; however, no difference was observed in VO_2_max values between vegetarian and omnivorous male athletes [58]. We need more high-level studies on the interaction between VO_2_max and vegetarian diet patterns in endurance athletes.

The availability of studies on vegetarian endurance athletes supports neither a positive nor a negative impact on exercise capacity [52,56]. Comparing the exercise capacity of lacto-ovo-vegetarian, vegan and omnivorous athletes, Nebl et al. [52] measured maximum power output (Pmax) during incremental exercise as the primary outcome of the study in determining exercise capacity, while maximum power output per lean body weight (Pmax_LBW_), blood lactate and glucose concentration during incremental exercise were evaluated as secondary outcomes. No differences were detected in Pmax, Pmax_LBW_, blood lactate and glucose concentrations between groups during increased exercise, suggesting that there was no difference in exercise capacity compared to the lacto-ovo-vegetarian (LOV), vegan or omnivorous diet pattern in endurance athletes [52]. In addition, a case study by Leischik and Spelsberg [56] assessed the exercise performance, cardiac status, and nutritional biomarkers of a male vegan ultra-triathlete and a control group of 10 Ironman triathletes during a Triple Iron ultra-triathlon (11.4 km swimming, 540 km cycling, and 126 km running). Apart from a mild thrombopenia with no pathological consequences in laboratory parameters, the vegan athlete did not have weakened nutritional biomarkers or impaired health symptoms. Additionally, the VO_2_max value of the vegan athlete was greater compared to the omnivorous athletes. Systolic and diastolic functions also did not differ between vegan and omnivorous athletes. The findings indicate that a well-planned vegan diet can provide adequate nutritional support for an ultra-triathlete [56].

In addition to these aforementioned benefits, vegetarian diets may also provide advantages for exercise capacity by increasing muscle glycogen levels [71], and delaying fatigue [74]. As for increasing glycogen stores, carbohydrate intake is considered the cornerstone of a better endurance performance by enhancing muscle glycogen stores, delaying fatigue, and providing athletes to compete at better and higher levels during prolonged periods [75]. Given the fact that the vegetarian diets are rich in carbohydrates (CHO) [71], such diets may offer more opportunities when considering races or training that can last at least six hours [2]. However, these data bring us to the point where foods high in CHO rather than diet types may be responsible for better performance. Taken together, both studies have shown that vegetarian diets neither benefit nor harm exercise capacity and endurance performance compared to omnivorous athletes. However, more studies are needed due to the small number of studies on the topic.

Studies have shown that the beneficial effects of vegetarian diets in alleviating oxidative stress and regulating the anti-inflammatory response are based on their enormous non-nutrient content called phytochemicals [4,76]. Polyphenols containing flavonoids, phenolic acids, lignans, and stilbenes are the most diverse non-nutrient group of phytochemicals that are produced as secondary metabolites throughout plants and have a broad spectrum of effects on metabolic health [77]. Polyphenol research of the athletic population has often been conducted using various fruits and vegetables, mainly berries [78], including blueberries [79,80,81,82], black currant [83], Montgomery cherry [84,85], and pomegranate [86]. Acute polyphenol intake or supplementation of ~300 mg 1–2 h before training or >1000 mg of polyphenol supplementation (equivalent to 450 g blueberries, 120 g blackcurrants or 300 g Montmorency cherries) 3 to more days (1–6 weeks) before and immediately after training is recommended as a countermeasure to improve antioxidant and anti-inflammatory response mechanisms [87]. However, only two studies examined the effect of vegetarian diets on exercise-induced oxidative stress in endurance athletes by comparing them with omnivorous diets, revealing contradictory results [53,55]. An incremental exercise test was applied in both studies. Nebl et al. [53] showed that nitric oxide levels, also known as an important biomarker for inflammation, endothelial and vascular function, did not alter between groups. In addition, exercise-induced malondialdehyde (MDA) concentration, an end product of lipid-peroxidation that is commonly measured to detect oxidative stress, significantly increased in vegan athletes in both studies, and in LOV athletes compared to omnivorous athletes [53]. Further, Potthast et al. [55] found a negative interaction between MDA, and sirtuin activities and antioxidant intakes such as ascorbate and tocopherol. These studies showed opposite results, against expectations, i.e., vegetarian diets increased the antioxidant response while suppressing the oxidant response. One explanation might be that the MDA test may not provide accurate measurement in biological samples due to its high reactivity and cross-reactions with other biochemicals available in the body despite its widely usage as an oxidative stress biomarker [88]. Therefore, studies with a greater sample size and including other oxidant parameters are needed to clarify these findings.

In addition to polyphenols, Interleukin 6 (IL-6) has often been identified as an inflammatory biomarker associated with fatigue, skeletal muscle inflammation, and differentiation of immune response, as well as an inducer of the metabolic acute phase response to infection [4,89,90,91]. It has been suggested that endurance athletes consuming vegetarian diets may have lower IL-6 concentrations and a lower IL-6 increase in response to endurance performance [4]. These data are explained by the positive interaction between muscle glycogen and IL-6 concentration, based on the information that higher muscle glycogen stores cause lower IL-6 elevations [92]. The higher CHO content of vegetarian diets may increase muscle glycogen stores, resulting in a down-regulated IL-6 response to endurance performance [4]. However, there are no data comparing the vegetarian and omnivorous diets for IL-6 concentration in endurance athletes.

One further point is the possible roles of vegetarian diets in URTI [73]. It is well known that endurance athletes are at greater risk for URTI due to prolonged and excessive training or races that cause immunosuppression and immune deficiency [93]. The possible link between URTI and a vegetarian diet may be explained with an emphasis on its polyphenolic content [94]. Polyphenol supplementation is also preferred in endurance athletes because of its debilitating role in URTI, one of the risk factors that often arise after immunosuppressive endurance exercise. A meta-analysis by Somerville et al. [73] reported that flavonoid supplementation reduced the incidence of URTI by 33% compared to a control group. Researchers also examined all factors that may cause a bias between studies, indicating that the risks for sequence generation, allocation concealment, and reporting bias are unclear in the included studies in the systematic review [73]. On the other hand, in a crossover design, Richter et al. [54] compared the influence of a 6 week LOV diet versus a meat-rich Western diet on in vitro measurements of immunologic parameters in male endurance athletes. The findings reported that no change was detected in CD3^+^ (pan T-cells), CD8^+^ (mainly T suppressor cells), CD4^+^ (mainly T helper cells), CD16^+^ (natural killer cells), CD14^+^ (monocytes) after the two diet trials and none of the immunological parameters differed from each other after the two diets. Studies have commonly focused more on diet content rather than diet pattern whether vegetarian or omnivorous. Therefore, the potential immunological benefits of vegetarian diets need to be investigated further.

A review investigating the effect of vegetarian diets on cardiovascular health in endurance athletes highlighted that vegetarian diets can provide better cardiovascular protection by reducing plasma lipid levels, exercise-induced oxidative stress, inflammation and blood pressure, and improving endothelial function and arterial flexibility [71]. One cross-sectional study confirmed the information by investigating the difference in heart morphology and function according to the vegan and omnivorous diets in amateur runners [59]. The results showed that vegans had better systolic function, determined by longitudinal strain (vegan:−20.5% vs. omnivore:−19.6%), and diastolic function in vegans, determined by higher E-wave velocities (87 cm/s vs. 78 cm/s), compared to omnivorous athletes [59]. Therefore, we can confirm that vegetarian diets may have a beneficial impact on cardiovascular function; however, we still need further investigation on endurance athletes.

Potential Risks of Vegetarian/Vegan Diets

Vegetarian and vegan diets offer several beneficial privileges for athletic populations [9,71]. However, the underlying mechanisms linking vegetarian diets to metabolic processes that may lead to undesirable effects on sports performance and, more importantly, metabolic health, should be considered beyond their beneficial functions [95]. In cases where athletes follow a vegetarian diet, issues related to the micronutrient deficiency, diet’s energy availability [96], relative energy deficiency syndrome (RED-S) [11], serum hormones [97,98], and protein quality/quantity [99,100] are topics that need to be addressed first.

Athletes who adhere to vegetarian diets are considered at high risk for deficiency of certain nutrients, especially when their dietary composition is not well-structured [10]. These risks are mainly due to the restriction of some food groups with a high nutrient density such as milk, meat, and eggs, the inability to access vegetarian foods when needed, or the development of early satiety and loss of appetite due to the high fiber content of vegetarian foods [95,101]. Furthermore, due to these dietary restrictions, athletes are at a higher risk for several micronutrient deficiencies including omega-3, iron, zinc, iodine, calcium, vitamin D, and vitamin B_12_ [101].

Nebl et al. [102] investigated the food consumption of vegan, lacto-ovo-vegetarian (LOV) and omnivorous (OMN) athletes according to the intake recommendations of the German, Austrian, and Swiss Nutrition Societies for the general population. Most athletes did not reach the recommended energy intake. Although omnivorous athletes consumed lower CHO compared to the recommended intake, vegetarian athletes consumed adequate amounts. For micronutrient intake, vegans achieved adequate iron levels by consuming only foods high in iron, while female LOV and OMN athletes achieved the recommended amount after supplementation. The results showed that all groups consumed enough of most nutrients. However, an analysis of the circulating state of nutrients is also needed to better interpret the effectiveness of dietary intake, particularly for vegetarian athletes [102]. A cross-sectional study by the same researchers [103] then compared the micronutrient consumption of LOV, vegan, and omnivorous recreational runners and found that 80% of each group had adequate vitamin B_12_ and vitamin D levels, and these parameters were higher in supplement users. Red blood cell folate exceeded the reference range; however, there was no difference in red blood cell folate among all groups [103]. No iron deficiency anemia was detected in any group, and less than 30% of each group were found to have depleted iron stores. The results suggest that a well-planned vegetarian diet can meet the athlete’s iron, vitamin D, and vitamin B_12_ needs [103]. These findings have been confirmed in case reports on vegan mountain bikers and ultra-triathletes [56,57]. Additionally, vegetarian diets are often inferior in quality compared to omnivores; this is due to anti-nutritional factors such as trypsin inhibitors, phytate, and tannins in those rich in vegetarian diets [104]. However, these challenges can be overcome by applying pre-cooking techniques described in detail in another review [105]. Therefore, it is obvious that vegetarian diets require careful monitoring in endurance athletes whose energy, macro and micronutrient needs are higher than their omnivore counterparts. However, with a well-planned diet and close monitoring, the nutritional needs of athletes can be successfully met, even ultra-endurance athletes.

Various metabolic risks such as iron deficiency anemia, menstrual disorders, musculoskeletal injuries, immunity, and hormonal irregularities occur in endurance athletes as a result of insufficient energy and nutrient intake following high-intensity endurance performance [106,107]. Relative energy deficiency syndrome has been found more often in vegetarian athletes, which causes endocrine and eating disorders that cause harmful diseases to metabolic health, reduces bone mineral density, and causes menstrual dysfunction [108,109]. Relative energy deficiency syndrome was developed to replace the Female Athlete Triad by broadening the definition to include male athletes and impaired physiological function caused by relative energy deficiency [109]. The key etiological factor of RED-S is a low energy availability which results in, but is not limited to, impairments of metabolic rate, menstrual function, bone health, immunity, protein synthesis and cardiovascular health [109]. In a study, researchers attribute this to either vegetarians’ food choices for low-energy-dense, high-fiber foods, even in high-energy situations, or restricted food intake behaviors by indicating dietary rules to mask vegetarians’ eating disorders [110]. Since low energy availability has already presented a challenging problem for endurance athletes independent of the diet pattern [111] and even healthy endurance athletes often cannot fully meet their body energy and vitamin requirements [112], nutritional adequacy and the quality of vegetarian diets are often questioned. However, studies examining the nutritional efficiency of vegetarian diets claimed the opposite results. Examining the diet adequacy and performance parameters of a vegan ultra-triathlete with 10 Ironman counterparts, a case report has revealed that a vegan athlete has no nutritional deficiencies or health disorders [56]. Researchers examined the spiroergometric, echocardiographic, or hematological parameters of a vegan ultra-endurance triathlete that has been vegetarian for 22 years and vegan for the past nine years. It has been found that a long-term vegetarian diet is not detrimental to metabolic health for a long-distance triathlete, even at micronutrient parameters associated with anemia. Although being a vegan athlete who consumes a well-planned diet does not have a detrimental impact in terms of cardiometabolic health and sports performance [56], findings need to be explored with a larger athletic cohort. These findings are similar to those of Wirnitzer et al. [57], who evaluated the food intake of a vegan mountain biker in the Transalp Challenge race (42 h). The researchers have highlighted that a carefully planned vegan diet strategy ensures that the race goals are achieved, and thus the race is completed in a healthy state [57]. Therefore, a well-planned vegan diet can be a great alternative for ultra-endurance athletes who endure extreme conditions such as psychological, physiological, endocrinological, and immunological stress-related metabolic challenges during prolonged training periods. In the last statement by the Academy of Nutrition and Dietetics on vegetarian diets, it was stated that vegetarian diets seemed more sustainable for all stages of life [113]. Researchers have suggested that well-planned vegetarian and vegan diets containing certain micronutrients such as high-quality plant protein, iron, n-3 fatty acids, Zn, Ca, iodine, vitamin B_12_, and vitamin D provide various health benefits regarding diseases such as hypertension, ischemic heart disease, diabetes and obesity [113]. In addition, given the content of vegetarian diets that can contain milk, eggs, or fish, vegetarian diets may be a better option for providing better nutritional density and quality than a vegan diet [99]. It is recommended that vegans carefully monitor blood vitamin B_12_ concentrations and supplement their diets, if necessary, with supplements or fortified foods [113]. Vegetarian and vegan nutrition programs should be planned by considering the above-mentioned data.

For many years, there have been claims that vegetarian diets negatively affect serum sex hormones [97,98], but data on the interaction between serum sex hormones and vegetarian diets remain controversial. In a crossover study conducted in 1992, Raben et al. [114] studied the effects of a 6 week lacto-ovo-vegetarian and omnivorous diet on serum sex hormones and endurance performance in eight endurance athletes. Although endurance performance did not differ according to the diet model, serum testosterone levels slightly decreased after six weeks of consuming a lacto-ovo- vegetarian diet. The researchers stated that these results may be related to dietary fiber binding to sex hormones and higher fiber intake in the lacto-ovo-vegetarian diet [114]. Considering the evidence in the literature that testosterone triggers muscle protein anabolism and lean body mass [115], a decrease in testosterone levels would cause an undesirable situation. However, a recent study in men from the national health and nutrition examination survey (NHANES) database, but not on athletes, found that a vegetarian diet did not link to serum testosterone levels [116]. Along with all data, the interpretation of the vegetarian diet as an attenuating factor to sex hormones by disregarding other confounding factors such as age, gender, training intensity, and emotional stress would be inappropriate [116] and needs further investigation.

The issue of the protein quality and quantity of vegetarian diets has long been controversial [99,117]. While some researchers note that vegetarian proteins have some missing specific amino acids [118], others state that including high-quality protein-rich foods such as legumes, seeds, nuts, and grains in a vegetarian diet is sufficient to meet the body’s amino acid requirement [119]. A vegan diet structure should be created by examining the protein content of the food consumed, especially in terms of quality and quantity. Determining the dietary protein quality using the Digestible Indispensable Amino Acid Score (DIAAS) method in omnivore and vegetarian athletes, Ciuris et al. [100] analyzed the diet content of 38 omnivore- and 22 vegetarian athletes. Vegetarian athletes had significantly lower lean body mass (LBM) compared to omnivores (−14%). Available protein was significantly correlated with strength (*r* = 0.314) and LBM (*r* = 0.541). The main findings revealed that vegetarian athletes needed to consume an additional 10 g of protein per day to achieve the recommended protein intake of 1.2 g·kg^−1^ body weight (BW) and an additional 22 g of protein to reach 1.4 g·kg^−1^·kg^−1^ BW [100]. Data on vegetarian proteins such as hemp, soy, potato, and rice proteins highlight that these vegetarian proteins contain sufficient high-quality protein content to increase muscle protein synthesis and post-workout recovery [119]. Rogerson [10] suggested that vegan athletes could improve their protein intake towards the higher limit of the International Society Of Sports Nutrition’s (ISSN) protein recommendation for athletes up to 2.0 g·kg^−1^ body mass per day. However, given that there is little evidence in the literature that vegetarian proteins are inadequate to provide an athlete’s needs or that vegetarian athletes need a higher protein intake [11,117], this recommendation needs further clarification with clinical research.

Additionally, the potential benefits of vegetarian diets are often attributed to their polyphenolic contents [4]. The intake of polyphenols with food may be the best choice for regulating body hormesis in the case of antioxidants due to the fact that polyphenol supplements may compromise the body’s antioxidant defense metabolism [87,120]. However, at this point, the bioavailability of polyphenols taken with food comes into question [121]. While some researchers have suggested that the recommended polyphenol intake can be achieved by consuming polyphenol-rich foods or as a polyphenol supplement [122], others claimed that some polyphenols, such as quercetin, cannot be taken naturally with food [123]. Keeping all this in mind, it is necessary to further clarify the possible mechanisms for how the bioavailability of polyphenols in the body and their effects on sports performance change with their consumption naturally.

With all the data obtained from studies, there is currently no certain evidence that omnivorous or vegetarian diets provide better metabolic health and performance benefits [52,53,55,56,57,59]. Therefore, more research is needed to clarify the optimal dietary recommendations for macro and micronutrients, as well as polyphenols, to maintain health and improve performance in endurance athletes following vegetarian diets.

### 3.2. High-Fat Diets

High-fat diets (HFD) have been widely applied for decades as a treatment option for certain diseases such as epilepsy or as an effective dietary strategy for weight loss [124]. In recent years, these diets have also become widespread in endurance athletes [14,15,16,17,18,19,21,22,23,24,38]. High-fat diets applied in the athletic population are grouped under two main categories: (1) a ketogenic low-CHO high-fat (K-LCHF) diet, and (2) non-ketogenic high-fat (NK-LCHF) diet (described in Table 2). While a ketogenic diet aims to increase blood ketone levels from 0.5 to 3.0 mmol/L, non-ketogenic diets aim to provide potential benefits without reaching higher blood ketone concentrations. Ketosis is considered as a survival mechanism for the body to equilibrate blood glucose during a metabolic crisis, such as a lack of calories or glucose, in fasting conditions, or prolonged exercise and to provide energy to the brain, whose survival depends on ketone body (KB) utilization in case of glucose deprivation [125].

In HFD studies on endurance athletes, K-LCHF diets have been commonly applied for diet periods ranging from three to 12 weeks [16,17,18,19,21,22,23,38]. Two studies, a case report (a 10 week K-LCHF diet) [21] and a cross-sectional study (a 20-month K-LCHF diet) [24], examined the effects of longer-term ketogenic diets on performance. For NK-LCHF diets, three studies, two crossover (a 2 week NK-LCHF diet) [26,28] and a cross-sectional (6-mth NK-LCHF diet) study [27], also investigated the impact of NK-LCHF diets on performance and lipoprotein profiles in endurance athletes (Detailed in Table 1). Besides these ketogenic diet applications, acute [30,31,32,33,34,35,36,37,38,39] or long-term [130] administration of KBs (in a ketone ester (KE) or ketone salt (KS) form) and CHO restoration following keto-adaptation [26,40,41,42,43,45,46,47] have also been evaluated in endurance athletes. Additionally, studies have been conducted to investigate the effects of an acute pre-exercise high-fat meal [51], and a short-term (1.5 days) fat supplementation during high-CHO diet administration [49,50]. In this section, we will discuss these high-fat studies in detail, with all their beneficial and harmful consequences for endurance athletes.

#### 3.2.1. Potential Beneficial Aspects of High-Fat Diets

High-fat diet administration has taken place in endurance athletes with the aim of improving the utilization of fatty acids and KB [14,19,20,24,25,26,28,32,33,34,35,36,41,42,43,45,46,47,49,50,51], sparing muscle glycogen stores [24,37,42,44,46,47], increasing weight loss, especially body fat mass [14,19,21,28], improving aerobic capacity [28], improving time to exhaustion [26,51] and time-trial performance [33,46,131], regulating performance-related parameters [34,36,39], increasing cognitive performance [38], regulating exercise-associated immunologic and hormonal response [15,22,30], increasing cellular gene expression [132], and attenuating overreaching syndrome [130].

One of the main goals of applying a high-fat diet to improve performance is to increase the body’s ability to use KB and fatty acids as an energy source [14,19,20,24,25,26,28,32,33,34,35,36,41,42,43,45,46,47,49,50,51]. The enhancement of the body’s ability to use KB as an energy source generally occurs in two type manipulations: (1) By restricting dietary CHO intake for a prolonged time, the body adapts metabolically to using KB instead of glucose; this process is called keto-adaptation [24]. (2) Acute KB supplementation instantly changes fuel usage from CHO to KB [30,31,32,33,34,35,36,37,38,39,130].

Improvement of fat utilization to fuel, especially during prolonged exercise, may provide advantages for endurance athletes, including the glucose-sparing effect that, in particular, has vital importance for the brain during times of glucose depletion [133]. While the intramuscular triglyceride stores are predominantly preferred to provide energy during low- to moderate-intensity exercise (50–75% VO_2_max), in moderate to vigorous-intensity exercises (>75% VO_2_max), muscle glycogen is used as the primary substrate to obtain energy provisions [134]. However, since the substrate utilization highly depends on the diet pattern, keto-adaptation results in a shift from glycogen to FFA or KBs, even during high-intensity exercises [21]. A number of studies such as K-LCHF [14,15,19,20,21,24,25] and NK-LCHF trials [26,28], acute KB administration [32,33,34,35,36,39], keto-adaptation followed by CHO loading [41,42,43,46], and pre-workout HF meal administration [51] proved that fat oxidation significantly increased at rest and during exercise after HFD applications. Only studies practicing the short-term fat administration during high-CHO diet administration in trained male cyclists revealed that overall fat oxidation did not alter during prolonged exercise and during submaximal or one hour time-trial (TT) exercise training [49,50]. However, one of the studies noted that fat oxidation significantly increased regardless of diet [50], while another highlighted that intramyocellular lipid utilization increased 3-fold in the fat supplemented group [49]. Taking all studies together, it seems that all applications aiming to increase fat ingestion provide better fat and KB utilization in the body, especially during exercise. This metabolic advantage appears to be unique for enhancing endurance performance.

However, along with the changes in substrate utilization towards fatty acids and KBs, KD might not be advantageous for exercise that highly relies on anaerobic metabolism and requires glucose flux such as short-duration exercise or long-duration exercise with interval sprints. In a randomized, crossover study in trained endurance athletes, it was stated that a 5 day fat adaptation followed by 1 day CHO restoration caused a decrease in glycogenolysis and PDH activation [47]. The findings suggested that this dietary manipulation could result in an increase in the NADH/NAD+ ratio or the Acetyl-CoA/CoA ratio, which could result in sustained attenuation of PDH activity and impaired glycolysis metabolism. Further research should be elucidated on the possible interaction between impaired glycolysis metabolism and ketogenic diets on prolonged exercise with anaerobic metabolism or high-intensity intermittent exercise.

As it is well known that depleting glycogen stores is one of the major causes of fatigue during endurance exercise [2], HFD also aims to reduce muscle glycogen utilization to ensure CHO availability for longer periods of time during endurance training. Although one study on endurance-trained male cyclists showed that muscle glycogen utilization significantly decreased after a 10 day fat adaptation followed by 3 day CHO restoration trial compared to a high-CHO trial [46], others investigating muscle glycogen utilization claimed that no difference was observed between the intervention and the control trial [24,37,42,47]. In addition, a cross-sectional study on male endurance runners stated that muscle glycogen utilization did not alter after an average of a 20-month K-LCHF or high-carbohydrate (high-CHO) diet. Therefore, studies on HFD and its “muscle glycogen sparing effect” remain controversial. We cannot conclude that HFD provides an advantage to spare muscle glycogen during endurance training. Further work is needed to assess muscle glycogen utilization.

K-LCHF diets might be an effective option for athletes who aim to lose body weight (BW) and body fat while sparing muscle mass [14,19,21,28]. A crossover study assessing the effects of a long-term (4 week) K-LCHF diet rich in polyunsaturated fatty acids on aerobic performance and exercise metabolism in trained off-road cyclists revealed that BW and body fat percentage decreased after long-term KD [28]. It was also stated that the long-term K-LCHF diet improved maximum oxygen consumption and decreased post-exercise muscle damage. The findings suggest that a long-term K-LCHF diet may provide advantages to both body composition and endurance performance. However, another study claimed that long-term KD (for 12 weeks) caused a decrease in both body fat percentage (5.2%) and body mass (5.9 kg) in endurance-trained athletes [14]. However, results also showed that although long-term KD resulted in improved body composition, it had no impact on 100 km TT performance. Consistent with this study, Heatherly et al. [19] investigated the impact of a 3 week ad libitum ketogenic diet on markers of endurance performance in recreationally competitive male runners. Results showed that the body composition of subjects positively changed with a decrease of ~2.5 kg BW and skinfold thickness occurring at multiple sites in the trunk region. However, KD did not affect exercise-induced cardiorespiratory, thermoregulatory, and perceptional responses and 5 km TT performance, and perceived exertion [19]. Findings indicate that KD may be an alternative strategy for reducing fat mass regardless of endurance performance.

On the other hand, Zinn et al. [21] investigated the 10 week ketogenic diet experiences of five endurance athletes and the effects of this diet on body composition and exercise performance. Although body mass and the sum of skinfolds were reduced by an average of 4 kg and 25.9 mm, respectively, endurance athletes experienced an inability to maintain high-intensity exercises during this period [21]. These findings raised doubts about the use of KD for weight loss in endurance athletes. In addition to that, a recent study compared the efficiency of two energy-reduced (−500 kcal·day^−1^) diets, including a cyclical ketogenic reduction diet (CKD), defined as a high-fat low-CHO (>30 g·day^−1^) diet for five days, followed by a high carb diet (8–10 g/body FFM) for two days, and a nutritionally balanced reduction diet (RD), a typical diet containing 55% CHO, 15% protein, and 30% fat, on body composition and endurance performance in healthy young males [135]. Results revealed that both diets reduced body weight and body fat mass. However, while CKD-related weight loss is due to decreased body fat, body water, and lean body mass, RD leads to a reduction in body weight mainly by reducing body fat mass [135]. Among all of these findings, one should note that adherence to a weight loss diet is major factor in achieving a target that does not significantly require KD consumption.

Several studies determined the potential impact of HFD on aerobic capacity [16,17,20,23,25,28,39]. It is well known that VO_2_max is referred to as a gold standard method to measure aerobic fitness [136]. Therefore, studies on KD, N-KD, and acute KE ingestion in endurance athletes stated that these diet manipulations had no effect on VO_2_max performance [16,17,20,23,25,39], except for a 4 week KD study on off-road cyclists by Zajac et al. [28]. Studies arguing that HFD was ineffective on aerobic capacity also showed that this HFD caused a decrease [16,23] or no change [25,39] in TT performance, and no alteration in time-to-exhaustion (TTE) performance [20]. Therefore, HFD seemed to fail to increase aerobic capacity and endurance in endurance athletes.

Researchers evaluated multiple performance-related factors such as TT performance [14,19,23,25,29,33,34,35,36,38,39,41,42,43,45,46,48,49,50], TTE performance [18,20,21,26,31,37,51], lactate concentration during exercise [33,34,36,39], and post-exercise muscle damage [28] to determine the effects of HFD on sports performance. While research on TT and TTE performance in endurance athletes revealed controversial results, the majority of the studies declared that no alterations were observed in TT [14,19,25,29,34,36,38,39,41,42,43,45,49,50] and TTE [18,20,26,31,37] performance after the HF-associated applications. Additionally, two well-controlled studies of Burke et al. [16,23] underlined that a 3.5 week K-LCHF diet not only decreased 10 km race walk performance, but also increased oxygen cost and perceived exertion throughout exercise. These findings suggest that HFD has no advantage or may even negatively affect exercise performance. However, some points should be taken into account when interpreting these findings. Five of eight studies on perceived fatigue during endurance performance revealed that no differences were detected between HFD and control trials [18,19,38,39,49]. Similar results were also observed in studies on lactate concentration during exercise [34,36,39]. It seems that HFD altered neither perceived exertion nor plasma lactate concentrations. Another important point for endurance performance is the maintenance of blood glucose concentration during exercise [38]. Changes in blood glucose levels during exercise were investigated in acute KB ingestion trials [34,36,38,39]. Three of four studies indicated that blood glucose concentrations were maintained during endurance exercise and were found to be similar between control groups [34,38,39].

Although these results are promising, blood glucose changes should also be examined in studies involving HFD manipulations. Additionally, a crossover study evaluating the efficiency of a 4 week NK-LCHF diet application on off-road cyclists stated that blood CK and LDH concentration, known as muscle damage biomarkers, significantly decreased at rest and during the 105 min exercise protocol in the NK-LCHF diet trial [28]. These findings also appear promising. It should be noted that studies reporting that TT or TTE performance did not change after HFD interpreted the study results based on statistical significance. It should also be noted that, although considered as statistically insignificant, a few minutes can be crucial in winning a race. Therefore, this point should be considered when interpreting the study results. Lastly, for post-exercise recovery, Volek et al. [24] indicated that long-term (at least 6 months) LCHF diets resulted in an increased fat oxidation rate and a higher peak exercise intensity in endurance athletes compared to counterparts consuming high-CHO low-fat diets. Moreover, although the LCHF diet group consumed 10% CHO, whereas the habitual high-CHO group consumed 59% CHO, there was no difference between the LCHF and high-CHO low-fat diets for 2 h post-exercise recovery [24]. These results suggest that long-term LCHF diets can improve post-exercise recovery, especially in ultra-endurance events where the glycogen-sparing effect and adequate post-exercise recovery are crucial for a better performance. Keeping all these findings in mind, although studies on TT and TTE performance mostly found no advantages of HFD or revealed controversial results, performance-related parameters may be positively affecting the HFD. More work is required to clarify this information.

Ketone body consumption in endurance athletes may increase endurance performance by up-regulating physiological parameters and increasing metabolic efficiency [126]. For instance, Cox et al. [33] conducted comprehensive research including five separate studies on the effect of ketone esters (KE) on the performance of 39 endurance athletes. Twenty minutes after consumption of the ketone ester-based drink, blood ketone concentrations rapidly increased to 2 mmol/L and remained high with a slight drop, reaching a new steady state approximately 30 min following subsequent exercise at 75% Wmax exercise intensity. Findings from the study showed that acute nutritional ketosis caused by the consumption of KE resulted in metabolic improvements in endurance performance by enhancing metabolic flexibility and energy efficiency, rapidly altering substrate utilization towards ketone bodies for oxidative respiration, sparing intramuscular BCAA concentration by reducing BCAA deamination, increasing muscle fat oxidation even though in the presence of glycogen, and decreasing blood lactate levels during exercise [33]. On the other hand, most of the studies (6 of 10) applying acute KB intake showed that this practice did not improve TT [34,36,38,39] and TTE performance [31,37]. Study findings remain unclear, and the impact of KB on exercise performance needs further clarification.

The efficacy of HFD on cognitive performance has been investigated in studies on acute KE [39] and KS [38] administration and fat-enriched feeding during high-CHO diet administration [50]. Prins et al. [38] administered one (22.1 g) or two (44.2 g) servings of KS or placebo to recreational male distance runners 60 min before a 5 km TT performance, and noted a possible dose–response interaction between KS supplementation and cognitive performance. On the other hand, studies including acute KE administration [39] and fat-enriched feeding during high-CHO diet administration [50] showed no alteration in cognitive performance. A study applying the high-CHO diet supplemented with fat on trained male cyclists highlighted that a possible explanation for this result is that the study protocol, including 1 h of fixed-task simulated TT performance, may not be sufficient to create mental fatigue [50]. However, the study on acute KE intake found similar results despite applying an exercise protocol (1 h submaximal exercise at 65% VO_2_max followed by a 10 km TT) that caused more fatigue [39]. Taken together, studies did not confirm the exact efficiency of HFD on cognitive performance and the interaction needs further investigation.

Few studies investigated the potential influence of HFD on immunologic and hormonal response in endurance athletes [15,22,30]. Assessing the impact of acute (2 day) and prolonged (2 week) adherence to an K-LCHF diet on exercise-induced cortisol, serum immunoglobulin A (s-IgA) responses in a randomized, crossover manner, researchers indicated that a lower cortisol response at week 2 was observed compared to day 2 in the K-LCHF trial (669 ± 243 nmol/L vs. 822 ± 215 nmol/L, respectively) [15]. However, a better exercise-induced cortisol response was found in the HCF trial at both day 2 and week 2 (609 ± 208 nmol/L and 555 ± 173 nmol/L, respectively). Additionally, no differences in s-IgA concentrations were observed at week 2 between the K-LCHF diet and high-CHO diet [15]. Another study by Shaw et al. [30] determined the impact of acute KE supplementation (R,S-1,3-butanediol (BD); 2×0.35 mg·kg^−1^ BW; 30 min before and 60 min after exercise) on the T-cell-associated cytokine gene expression within stimulated peripheral blood mononuclear cells (PBMC) following prolonged, strenuous exercise in trained male cyclists. No alteration was detected in serum cortisol, total leukocyte and lymphocyte, and T-cell subset levels, IL-4 and IL-10 mRNA expression, and the IFN-γ/IL-4 mRNA expression ratio between the KE and placebo trials during exercise and recovery. However, a transient increase was observed in T-cell-related IFN-γ mRNA expression throughout exercise and recovery in the KE trial. Results indicated that acute KE supplementation may provide enhanced type-I T-cell immunity at the gene level [30]. The same researchers investigated the potential effect of a 4.5 week K-LCHF diet on resting and post-exercise immune biomarkers in endurance-trained male athletes in a randomized, repeated-measures, crossover manner [22]. T-cell-related IFN-γ mRNA expression and the IFN-γ/IL-4 mRNA expression ratio within multiantigen-stimulated PBMCs were greater in the K-LCHF trial compared to the high-CHO trial. Furthermore, a significant rise was observed in the multiantigen-stimulated whole-blood IL-10 production, an anti-inflammatory cytokine, post-exercise in the K-LCHF trial. The results indicated that a 4.5 week K-LCHF diet caused an increase in both pro- and anti-inflammatory T-cell-related cytokine response to a multiantigen in vitro [22]. Keeping the studies on immunologic and hormonal response to HFD in mind, although post-exercise pro- and anti-inflammatory T-cell-related cytokine response alters after a K-LCHF diet or acute KE supplementation, it remains uncertain how these alterations influence the immunoregulatory response. Therefore, more work is required to elucidate the interaction by adding clinical illness follow-up and tracking immunomodulatory metabolites using metabolomic approaches.

Antioxidant specialties of HFD may be discussed on the basis of KB [124]. Antioxidant activity of KBs is one of the multidimensional properties that determine their metabolic activity in the body. The main potential antioxidant properties of KB are mainly explained by its effects on neuroprotection, inhibiting lipid peroxidation and protein oxidation, and improving mitochondrial respiration [137]. However, as there is no study investigating the impact of KB on exercise-induced oxidative stress in endurance athletes and the evidence on the impact of KB on exercise-induced oxidative stress is limited, future studies in this field are needed.

Another therapeutic benefit of KD may be linked to increased Fibroblast Growth Factor 21 (FGF21) [132]. Fibroblast Growth Factor 21 acts as the primary regulator of skeletal muscle keto-adaptation by increasing activation of the AMP-activated protein kinase (AMPK)—sirtuins 1 (SIRT1)—peroxisome proliferator-activated receptor coactivator 1 (PGC-1) pathway, resulting in increased mitochondrial biogenesis, development of IMTGs, and ketolytic gene expression [138]. However, in a study on 5-d fat adaptation followed by 1-d CHO restoration, a significant decrease was observed in the exercise-induced AMPK-1 and AMPK-2 activity in the fat-adapted trial despite the higher AMPK-1 and AMPK-2 activity before exercise. Therefore, more work is required to interpret the possible interaction accurately.

Ketone bodies may have a particular metabolic advantage, not only providing a source of oxidizable carbon to maintain energy needs but also acting as a potential regulator of overtraining by directly regulating autonomic neural output and inflammation [139,140]. One study applying three weeks of KE intake during prolonged extreme endurance training investigated the effects of KE on overreaching symptoms [130]. Ketone ester ingestion significantly increased sustainable training load (15% higher than the control group), and prevented the increase in nocturnal adrenaline and noradrenaline excretion induced by strenuous training [130]. These findings suggest that KE supplementation during exercise substantially reduces the development of overreaching, which is a detrimental factor for endurance performance. In addition, growth differentiation factor (GDF-15), an established biomarker for nutritional and cellular stress, increased 2-fold less in the KE group than the control group. However, this study was conducted on healthy, physically active males, and it is not exactly known whether the same effects can be achieved in endurance athletes [130]. For this reason, it is necessary to examine the same mechanism, especially on endurance athletes with intense and frequent training periods.

#### 3.2.2. Potential Risks Regarding High-Fat Diets

Some researchers have also investigated HFD’s potential risks on endurance, including an increased oxygen cost and an impaired running economy [16,23], an altered blood acid-base status [17,31], compromised gastrointestinal (GI) symptoms [32,34,35,37,48], reduced bone formation markers [40], increased cholesterol and lipoprotein levels [27], a decreased appetite [37], and thereby worsened performance.

The deterioration of the running economy and increased oxygen cost during endurance exercise are considered to be major potential disadvantages of HFD. Burke et al. [16,23] demonstrated with two separate studies in elite race-walkers that a 3 week K-LCHF diet during intensity training impaired endurance performance by decreasing exercise economy, which has vital importance in endurance performance, despite enhancing peak aerobic capacity (VO_2_peak). Another study by Burke et al. claimed that although KD elevated glycogen availability, it still impaired endurance performance mainly by blunting the CHO oxidation rate [141]. In addition, LCHF diets can also impair endurance performance by increasing perceived fatigue [15,16,23]. The reason why K- LCHF diets cause increased fatigue is thought to be a gradual increase in non-esterified fatty acids (NEFAs) with the LCHF diet [142]. Non-esterified fatty acids compete with the tryptophan, a neurotransmitter highly associated with the central fatigue, for binding to albumin, thus resulting in an increase in free tryptophan transfer from the blood–brain barrier towards the brain. However, as we discussed above, the majority of studies found no alteration in perceived exertion during endurance performance [18,19,38,39,49].

Studies on well-trained endurance athletes revealed that neither keto-adaptation nor CHO restoration followed by keto-adaptation improves endurance performance, especially at multistage ultra-endurance events with intermittent sprints [42,45]. For instance, investigating the impact of a 6 days high-fat (68% fat) diet followed by 1 day CHO loading or high-CHO diet (68% CHO) for seven days on performance parameters during the 100 km time trial, Havemann et al. [45] found that 100 km time trial performance assessed by heart rate, perceived exertion, and muscle recruitment did not differ between groups; however, the 1-km sprint power output decreased more in the high-fat diet group than in high-CHO counterparts. Although an improvement was expected in high-intensity sprint bouts after an NK-LCHF diet due to its sparing effect on muscle glycogen, the findings revealed the opposite, decreasing the high-intensity sprint performance, a crucial parameter for endurance performance [45]. On the contrary, McSwiney et al. [14] also evaluated the impact of K-LCHF diets on 100 km TT performance and 6 s sprint peak power, indicating that although TT performance did not differ between the K-LCHF diet and high-CHO diet groups, 6 s sprint peak power significantly increased (+0.8 W·kg^−1^ rise) compared to the high-CHO group (−0.7 W·kg^−1^ decrease). More research is required to clarify these contradictory results.

Maintaining the acid-base balance in the body during exercise, especially during strenuous exercise, is important to delay acidosis and fatigue and thus to maintain endurance performance [143]. Exercise is a well-known factor that alters the acid-base state [143]. In addition to exercise, the macronutrient composition of dietary patterns can also affect acid-base balance and systemic pH and HCO_3_ levels [31]. Some researchers claimed that HFD can alter circulating acidity by increasing acidic KB circulation in the body [144], while others state that acid-base balance can be well regulated by improving the adaptive mechanisms, regardless of diet [17]. The potential effect of HFD on blood acid-base status, blood pH, and HCO_3_ concentrations was evaluated in only two studies of endurance athletes. The potential effect of HFD on blood acid-base status, blood pH, and HCO_3_ concentrations was evaluated in two studies, one evaluating a 3 week ketogenic diet [17] and the other an acute KE intake in endurance athletes [31]. The study findings showed that neither K-LCHF diet nor acute KE intake affected blood pH and HCO_3_ status and acid-base status [17,31]. One explanation is that both studies included well-trained endurance athletes. It is suggested that well-trained athletes can regulate the body acid-base balance well regardless of the diet by developing a metabolic adaptation to strenuous exercise. Therefore, the potential effect of HFD on acid-base status can be interpreted as negligible when applied to well-trained endurance athletes.

Gastrointestinal symptoms triggered by an HFD have commonly been seen during KB consumption [32,34,35,37]. A study investigating the kinetics, safety and tolerability of KB revealed that ketone esters may only cause GI symptoms when high doses (2.1 g·kg^−1^) are consumed [145]. However, although studies administered a low-dose KE in endurance athletes, the findings stated that acute KE ingestion caused an increase in low to severe GI symptoms, including nausea, reflux, dizziness, euphoria, and upper-abdominal discomfort [32,34,35,37]. One study by Dearlove et al. [32] compared the dose–response interaction between acute low- or high-dose KE ingestion (0.252 g·kg^−1^ vs. 0.75 g·kg^−1^, respectively) and GI symptoms. Findings showed that no GI discomfort was observed in the low-dose KE ingestion, while nausea symptoms were elevated in the high-dose KE trial. Although the high dose administered in this study (0.75 g·kg^−1^) remained much lower than the high dose (2.1 g·kg^−1^) that was claimed to cause GI symptoms, it still caused exercise-induced nausea in endurance athletes. In addition, Mujika [48] investigated the race performance and GI symptoms of a LOV male endurance athlete who adhered to an LCHF diet for 32 weeks. The athlete participated in three professional races while on the LCHF diet in weeks 21, 24, and 32. Although he suffered worse race experiences on the LCHF diet, no alteration was observed in GI symptoms. This result may be due to the athlete’s adaptation to the ketogenic diet [48]. Taken together, while long-term keto-adaptation may inhibit the increase in GI symptoms, it should be taken into account when applying to endurance athletes that acute KE intake may be disadvantageous on exercise-induced GI symptoms. Interestingly, Zinn et al. [21] showed that endurance athletes suffered from constipation during the diet application after a 10 week K-LCHF diet, which might be important for the gut microbiome and well-being. This possibility may also be kept in mind while applying a ketogenic diet. In case of a similar situation, fiber and water intake should be calculated and closely monitored to eliminate constipation-associated problems.

Another less-studied potential disadvantage of HFD is its potential impact on decreasing appetite [37], bone formation markers [40], and increasing cholesterol and lipoprotein profile [27]. A randomized, crossover study evaluating the effects of acute KE ingestion early in a cycling race on glycogen degradation in highly trained cyclists showed a significant attenuation in the perception of hunger, determined using a validated 10-point visual analog scale [37]. This potential effect of HFD on appetite should be taken into account, especially during HFD administration planned for long-term application.

Heikura et al. [40] investigated the effects of a 3.5 week K-LCHF diet followed by CHO restoration on bone biomarkers in male and female race walkers. Their findings showed a meaningful increase in bone resorption markers at rest and post-exercise while a significant attenuation in bone formation markers at rest and throughout exercise in K-LCHF diet trial occurred. However, these alterations partially recovered after CHO restoration [40]. As only one study investigated the interaction between bone markers and ketogenic diets in endurance athletes, and a recent narrative review on ketogenic diets and bone health noted that we do not have enough high-quality experimental research to adequately clarify the potential disadvantages of ketogenic diets on bone health, we need more high-quality research on this topic.

Only one cross-sectional study of 20 competitive ultra-endurance athletes investigated the interaction between a long-term low-CHO diet and the circulating lipoprotein and cholesterol profiles [27]. Although a higher level of exercise tended to lower total and LDL-C concentrations, a hypercholesterolemic profile was observed in ultra-endurance athletes who adhered to a low-CHO diet, suggesting that a possible explanation may involve an expansion of the endogenous cholesterol pool during keto-adaptation and may remain higher on a low-CHO diet. Further, a higher consumption of saturated fat (86 vs. 21 g·day^−1^) and cholesterol (844 vs. 251 mg·day^−1^), and lower fiber intake (23 vs. 57 g·day^−1^) may be another cause of these hypercholesterolemic profiles of ultra-endurance athletes [27]. However, due to the small sample size (*n* = 20) and the lack of checking for familial hypercholesterolemia or specific polymorphisms [27], future work is needed to evaluate this interaction in depth.

Another possible pathway is that KD high in protein causes an increase in ammonia, thereby altering both brain energy metabolism and neuronal pathways, thus triggering central fatigue [146]. Both NEFA and ammonia may lead to increased central fatigue during exercise in endurance athletes adopting KD [142]. The interaction between the gut–brain axis can have critical importance to reveal performance- and, especially, fatigue-related metabolism during endurance events [147]. However, none of the HFD studies on endurance athletes studied the gut–brain axis, increased ammonia concentration, or endurance performance. Another point regarding a high protein intake during KD is that a high protein consumption can disrupt ketosis by providing gluconeogenic precursors, thus inducing gluconeogenesis [148]. Therefore, moderate protein consumption is generally recommended during KDs. As we know that endurance athletes tend to consume more protein intake (1.2–2.0 g·kg^−1^ BW·day^−1^) [149], this important effect of protein on ketosis should be kept in mind during the KD administration periods.

There are some important points that need to be considered before applying an HFD in endurance athletes. During NK-LCHF diet applications, the metabolic adaptation of muscle may evolve towards oxidation of fat as the primary energy source (maximum fat oxidation rate (fat max) from 0.4–0.6 g·min^−1^ to 1.2–1.3 g·min^−1^) [139]. However, glycogen stores may not provide enough glucose to power the brain, thus increasing fatigue [150] and decreasing endurance performance. For this reason, the adaptation period should be chosen carefully in order to alleviate the side effects of transition periods. Phinney et al. [20] noted that ketogenic high-fat diets may impair performance at first (a reduction of approximately 20%), but improvements in performance (up to a 155% increase) can be observed after metabolic adaptation to the ketogenic state.

Another important point that needs to be considered while planning further studies on HFD is to evaluate blood ketone concentration at frequent intervals during the study application period [151]. A review investigating the role of ketone bodies on physical performance found that 7 out of 10 studies included in the review failed to reach BOHB concentrations at the 2 mmol/L threshold, but only caused an acute ketosis state (B-OHB > 0.5 mmol/L) [151]. Another significant point is which KB type should be used [152]. The impact of ketone bodies on metabolism differs according to the type (ester-based form or salt-based form), and optical isoform (e.g., L or D isoforms of BOHB) consumed [137]. For example, D-βOHB is produced from acetoacetate (AcAc), released by the liver, and is actively used in metabolic pathways [153], while L-OHB is an intracellular metabolite known for having less activity in oxidative metabolism [150]. Therefore, L-βOHB supplementation may not provide the performance-related benefits of ketone bodies. These results explain that the specific effect of KD or KB on physical performance awaits further investigation, as most studies of KB failed to achieve the required ketone concentrations or applied ineffective KB to enhance endurance performance [152].

To conclude, there are several HFD strategies, as discussed in detail above, practiced by endurance athletes. However, while these diets may provide performance and health benefits, they are sometimes not effective at all or create many problems for endurance athletes. In addition, the physiological response to acute (exogenous) or endogenous nutritional ketosis may vary between highly trained endurance athletes and untrained individuals [140]. Therefore, it should be noted that these strategies may not be suitable for all endurance athletes. At first glance, while high-fat diets may seem like a promising approach to endurance performance, more research is needed to keep in mind all study results.

### 3.3. Intermittent Fasting

Intermittent fasting (IF) is defined as a period of voluntary withdrawal from food and beverages. It is an ancient approach that is implemented in different formats by different populations around the world [154]. Intermittent fasting diets have become more prevalent in recent years, including the scientific literature investigating the metabolic interaction between IF and health, as well as in the media and among the public [127]. Intermittent fasting diets are divided into four groups: (1) complete alternate-day fasting, (2) modified fasting, (3) time-restrictive eating and (4) religious fasting such as Ramadan IF (R-IF) (explained in detail in Table 2) [127].

#### Intermittent Fasting and Sports Performance

##### Possible Benefits of Intermittent Fasting in Endurance Athletes

Studies on IF in endurance athletes have often been conducted during the religious fasting period (R-IF) [60,61,63,64], with few studies investigating the effects of time-restrictive eating (16:8) on endurance performance and health-related effects [62,65]. Fasting diets may alter metabolic pathways in the body by acting as a potential physiological stimulus for ketogenesis [155], regulating metabolic, hormonal and inflammatory responses [61], and stimulating mitochondrial biogenesis and suppressing mTOR activity [155], and regulating body composition [62,65].

Energy restriction/fasting for more than 12 to 16 h leads to a metabolic switch in basic energy fuels from carbohydrates to fats, resulting in metabolic ketosis, the same as the ketogenic diets [155]. These KD-like alterations in substrate uses are believed to serve as an inductor for fat oxidation, and a preservative for muscle mass and function [156].

The effect of fasting diets on muscle cells is generally known to be similar to aerobic exercise, including stimulation of mitochondrial biogenesis and suppression of mTOR activity [157]. However, the main mechanism on fasting diets is driven by fatty acid metabolism and peroxisome proliferator-activated receptor delta (PPAR-d), instead of Ca^2+^, which is known to be effective in aerobic exercise [155]. Although the main mechanism on muscle cells differs between exercise and fasting diets, research findings suggest that application of a fasting diet along with exercise could switch cellular metabolism from glucose to ketone bodies [156], thereby inducing ketone utilization, which might, in turn, trigger mitochondrial biogenesis and preserve muscle mass [158]. Although the potential benefits of IF on mitochondrial biogenesis and mTOR activity appear promising, no study has investigated these metabolic interactions in endurance athletes adhering to IF.

The impact of R-IF on hormonal, metabolic and inflammatory responses is a less-studied point in terms of IF diets. In a study on middle-distance runners, Chennaoui et al. [61] examined the effects on R-IF on the hormonal, metabolic and inflammatory responses in a pre–post-test study design. Researchers applied a maximal aerobic velocity test 5 days before, 7 and 21 days after Ramadan. No change was observed in the testosterone/cortisol ratio during the RIF trial. A significant rise was reported in IL-6, adrenaline, and noradrenaline concentrations after the RIF; however, all parameters returned to baseline levels 7 days after exercise [61]. More work is needed to interpret these results effectively.

Another aspect of IF is its impact on the body composition of endurance athletes. Studies on endurance athletes and TRE (16:8) revealed that TRE caused a meaningful decrease in BW and body fat percentage in endurance athletes [62,65]. Moro et al. [62] claimed that although VO_2_max and endurance performance did not change after a 4 week TRE, a meaningful rise in the peak power output/BW ratio was due to the BW loss. However, another study showed a decrease in TT performance (−25%) and no improvement in running efficiency after R-IF in well-trained middle-distance runners [65]. Taking these studies into account, although IF may provide some benefits by decreasing BW and body fat percentage, we cannot assume that it positively affects endurance performance.

##### Risks to Be Considered When Applying Fasting Diets

Potential risks of IF diets are reduced endurance capacity [60], increased fatigue [61,63], altered sleep habits (i.e., delayed bedtime, decreased sleep time) [61,63,64], and dehydration [159] in endurance athletes.

Studies on IF diets and endurance capacity and performance-related parameters have produced conflicting results in endurance athletes [60,62,64]. Both R-IF and TRE studies on endurance athletes stated that IF diets had no influence on the aerobic capacity, determined by VO_2_max [60,62,64]. Additionally, one study on TT performance and R-IF in well-trained middle-distance runners showed that R-IF caused a decrease in TT performance [60]. However, another study determining the impact of the CHO mouth rising technique on 10 km TT performance declared that the CHO mouth rising technique provided benefits by increasing 10 km TT performance [64]. For TRE and endurance performance, Moro et al. [62] revealed that a 4 week TRE had no impact on endurance performance. As for evaluating performance-related parameters, several researchers investigated the exercise-induced fatigue, blood lactate, glucose, and insulin concentrations in endurance athletes [61,63,64,65]. Exercise-induced fatigue, as determined by the Fatigue score [61] and the Rated Perceived Exertion (RPE) Scale [63], increased after a maximum aerobic speed test and an intensive endurance training, while it decreased significantly in an R-IF trial applying mouth rising during a 10 km TT performance [64]. One TRE study also showed that blood lactate, glucose and insulin concentrations did not alter during an incremental test [65]. We know that endurance exercise lasts more than an incremental test duration. Therefore, although blood parameters were well-maintained during an incremental test, we cannot interpret the study as the parameters will be preserved during prolonged strenuous exercise. Since there are few studies on endurance performance and IF, further studies should be conducted with an exercise protocol similar to races and competitions, including all performance-related parameters.

One study assessed the effect of R-IF on cognitive function in a non-randomized, repeated-measures, experimental design manner [63]. No difference was observed in cognitive performance, measured using reaction time and mean latency times on simple and complex tasks during Ramadan in trained male cyclists. Therefore, the implementation of IF diets to increase endurance capacity, improve performance-related parameters or cognitive performance does not appear to be a well-approved strategy. On the other hand, it would be wrong to refer the IF diet as a detrimental strategy due to the controversial findings of studies. Further, a review of the role of R-IF in sports performance, which included well-controlled studies, reported that although R-IF generally affected athletic performance with a few declines in physical fitness at a modest level, including perceived exertion, feelings of fatigue, and mood fluctuations, these negative effects may not cause a decrease in sports performance [160]. Furthermore, while prolonged fasting has detrimental effects on endurance performance by decreasing endurance time and causing carbohydrate depletion, hyperthermia, and severe dehydration [161,162,163,164], IF causes preventable adverse effects on performance [160].

An important factor among the difficulties that IF can cause is the alterations in sleeping habits of endurance athletes who practice R-IF [61,63,64]. During R-IF, in contrary to other IF diets, sleeping periods alter due to the difference of fasting/feeding cycle, thereby disturbing the circadian sleep/waking rhythm [160]. These changes may trigger general fatigue, mood, and mental and physical performance in endurance athletes. A study on 8 middle-distance athletes who maintained training during Ramadan revealed that R-IF affected physical performance by disturbing sleeping habits, creating energy deficiency, and fatigue [61]. Another study on cyclists showed a significant reduction in the duration of deep and REM sleep two weeks after starting R-IF, although total sleep time was unchanged [63]. On the other hand, a study on adolescent cyclists also reported no change in total sleep time following R-IF [64]. As sleep is one of the major components for maintaining metabolic health and performance [165], during Ramadan IF, the sleep cycle of endurance athletes should be carefully monitored and effective sleep strategies should be developed for this period. Further, in order to determine the effects of Ramadan IF on sleep patterns more accurately, more objective sleep measurements should be applied.

Another adverse effect of R-IF on endurance athletes is the deterioration of hydration conditions before, during and after exercise [159]. Starting competition in euhydrated state is one of the key factors for greater performance [166]. Further, providing adequate fluid ingestion during exercise, especially prolonged strenuous training, has a major impact on body fluid homeostasis. Although glycogen breakdown provides an average of 1.2 L water [155], it is still not enough to meet the body fluid need during the marathon, especially in hot weather conditions [156]. Therefore, fasting due to lack of water/fluid consumption can create adverse health problems beyond performance detriments [146]. Although TRE diets allow the consumption of water and unsweetened coffee and tea, R-IF has restrictive rules that forbid the consumption of anything during the fasting state [127]. Therefore, the water balance and fluid strategy of endurance athletes should be carefully planned, especially for endurance athletes applying R-IF diets.

The adverse effects of the IF diets also vary according to the weather conditions during fasting, training severity, training load, and training level of athletes [159]. These factors and, more importantly, endurance athletes’ ability to cope with these metabolic changes determines how their sports performance will be during Ramadan. Evidence suggests that the performance success of athletes following an IF diet depends on their energy availability and macro and micronutrient intake, as well as training load and sleep length and quality [167]. Chennaoui et al. [61] suggested that athletes struggling with R-IF can reduce the negative effects of IF by reducing their training load and taking daytime naps.

Taking all studies into account, the efficiency of IF to improve exercise capacity and performance-related parameters still remains uncertain. Therefore, as we consistently repeat in the review, more work is needed before recommending these diets, especially in hot environments or during intense training periods. Since many Muslim athletes follow a month-long R-IF diet for religious reasons, even if there is a major competition or tournament [160], we need to develop effective strategies to maintain endurance performance and inhibit any decrease in endurance capacity during Ramadan.

### 3.4. Gluten-Free Diet

Exercise-induced GI symptoms in endurance athletes share common characteristics with Irritable Bowel Syndrome (IBS), including altered bowel functions (e.g., diarrhea, constipation), bloating, intestinal cramps, urge to defecate, and flatulence without any known organic disease [168]. These symptoms strongly affect the quality of life, psychological well-being, and also have quite a detrimental influence on exercise performance [1,168]. Therefore, several therapies have been developed for manipulating and attenuating these GI symptoms [169]. While drug-based treatments can be of benefit, certain foods are thought to trigger GI symptoms. In a research study, 63% of patients with IBS reported that some foods trigger their IBS symptoms [170]. Therefore, diet therapies gain more interest than other therapy options in patients with IBS and endurance athletes with GI symptoms. For example, a gluten-free diet (GFD) [128] and a low Fermentable Oligo-, Di-, Mono-saccharides, and Polyols (FODMAP) diet [171] are classified as elimination diets that both exclude or limit certain foods or nutrients that may cause undesirable GI problems such as abdominal bloating, cramps, flatulence or urge to defecate.

#### 3.4.1. Why Do Endurance Athletes Consider a Gluten-Free Diet to Be Beneficial?

A gluten-free diet is a strict elimination diet that requires the complete exclusion of gluten, a storage protein found in wheat, rye, barley seeds, and includes gluten-free foods and food products that do not contain gluten or have a gluten content of less than 20 ppm, as per European legislation [172]. It has been used for decades as a treatment for celiac disease (CD) or to treat other gluten-related disorders that require strict gluten elimination from the diet [173]. However, recently, gluten has been considered to be an inducer that triggers the pathophysiology of various conditions. Based on this theory, endurance athletes have widely practiced GFD even if CD or non-celiac gluten sensitivity (NCGS) has not been diagnosed [7]. Although they applied GFD as a possible dietary therapy because of their belief in a diet that could improve metabolic health and performance or alleviate exercise-induced GI symptoms, the results show no significant improvement in performance with GFD in non-celiac athletes [129].

A study of 910 athletes (male = 377, female = 528, no gender selected = 5) found that 41% of the athletes reduced their gluten consumption by approximately 50% to 100% due to their belief that gluten causes GI symptoms, inflammation, and decreased performance [7]. Endurance athletes in particular (70%) tend to exclude gluten from their diet. Almost half of the athletes who consumed GFD reported that at least one of their GI symptoms was attenuated with ongoing GFD [7]. Inconsistent with the study, a randomized controlled, double-blind, crossover study of 13 endurance cyclists with no known gluten-related disease who followed GFD or gluten-containing diet for a short period (7 days) showed that gluten elimination did not alleviate GI symptoms [66]. Additionally, neither plasma intestinal fatty acid binding protein (I-FABP), a marker of intestinal damage, nor TT performance differed between the groups. This is the only randomized-controlled study investigating the influence of GFD vs. gluten-containing diet on endurance performance and intestinal injury, and perceived well-being in endurance athletes [66]. Further research is required to elucidate the GFD, endurance performance and GI symptoms.

The best technique for identifying gluten-related issues is to remove gluten from the diet and check it for health effects in clinical practice [174]. With this gluten-related practice, athletes often self-diagnose that they have gluten-related disorders, resulting in gluten being excluded from the diet [129]. Assessing the presence of celiac symptoms, prevalence, and comorbidities in 141 collegiate athletes, Leone et al. [175] found that athletes reported being 3.85 times more likely to be diagnosed with CD and 18.36 times more likely to be associated with CD than the general population. This close association negatively alters the athlete’ health, leading to several detrimental consequences, including higher depression and perceived stress levels [175]. A possible explanation is that CD can be diagnosed faster as athletes monitor their health on a regular basis and work with an interdisciplinary team. The rapid detection of CD can provide an advantage to begin treatment as soon as possible, thereby reducing other harmful consequences associated with celiac disease.

A study on endurance athletes showed that they generally believed in GFD and its benefits to GI stress and exercise performance [176]. It is well known that the “belief effect” in athletes is an influential factor that can increase sports performance by 1 to 3% [177]. Whether gluten triggers exercise-related GI symptoms or whether endurance athletes with GI issues have a higher rate of NCGS remains unclear [66]. Additionally, switching to GFD can cause some healthy dietary changes in athletes, such as increased consumption of fruits, vegetables, legumes, and whole grains, and these changes may have more significant benefits on the GFD than gluten elimination [96]. Therefore, the gluten-free diet should not be recommended to non-celiac athletes (NCAs), as there is no evidence in the literature about its benefits to GI stress, immune response, and athletic performance [8,66].

#### 3.4.2. Possible Risks of a Gluten-Free Diet

The main concerns of GFD for endurance athletes can be classified as low energy availability [96] and the potential to create an energy deficit, micronutrients and fiber, leading to the RED-S [3]. Although GFD limits the consumption of certain gluten-containing foods rich in CHO that could lead to an energy deficiency [173], there is insufficient data to investigate the effect of GFD on energy deficiency in endurance athletes. We recommend that more studies are required on this topic, especially with a well-planned GFD for endurance athletes.

In addition, athletes consuming GFD need to greatly consider their diet as they need to control all foods for gluten content, which can negatively affect psychology [128]. For athletes with CD or other gluten-related clinical conditions, removing gluten from the diet is the only effective treatment [173]. In endurance athletes with CD, an increase in exercise performance and a decrease in GI problems were found after a gluten-free diet was adopted [178]. However, it is worth noting that endurance athletes need more energy to perform better in prolonged training and races, and gluten is present in carbohydrate-rich foods, which are the primary common source to meet their energy needs [112]. Gluten-free products are also known for their high cost and can sometimes be difficult to find [128]. Therefore, dietary gluten elimination may be an effective strategy for athletes with CD [173]. However, when applied to non-celiac athletes, it can create a large energy deficit and low energy availability, impairing both metabolic health and performance.

### 3.5. Low-FODMAP Diet

Exercise-related GI problems affect performance and health conditions in approximately 70% of endurance athletes [179]. Several foods are believed to trigger these GI symptoms, including foods high in fructose, lactose, digestible fibers, and undigested fermentable carbohydrates such as inulin and oligofructose, named “prebiotics” [180]. These fermentable short-chain carbohydrates are classified as FODMAP, including animal milk (lactose), legumes (galactooligosaccharides; GOS), wheat (fructans), fruits (high in fructose), and prebiotic foods (high in inulin, fructooligosaccharides (FOS) and oligofructose) [180,181]. Prebiotics are known for their beneficial effects on health, including reducing disease risks by increasing the microbial abundance of beneficial bacteria such as Bifidobacterium and butyrate producers [182]. However, they reach the colon and are fermented by colonic bacteria [183]. Thus, they can cause GI symptoms such as abdominal distress, bloating and gas, resulting in gas production, including hydrogen and methane and osmotic water translocation [184]. As a result, luminal distention and GI symptoms such as bloating, and cramps, can increase, impairing well-being and athletic performance [185]. Therefore, endurance athletes tend to remove high-FODMAP foods from their diets to eliminate their undesirable effects on the GI system [67]. In endurance athletes with exercise-induced GI symptoms, low-FODMAP diets could apply in two different processes, including the long or short term (both described in detail in Table 2) [8].

#### 3.5.1. Several Points Indicating That a Low-FODMAP Diet Is Advantageous

Endurance athletes’ expectations of a low-FODMAP diet are the same as those they have of GFD, including reduced GI symptoms, and thereby increased performance [8]. It is estimated that approximately 22% of endurance athletes have IBS [186]. Exercise-induced oxidative stress and physiological changes in the body can lead to impaired GI motility and intestinal permeability, which also occur as a result of IBS [147]. Foods rich in FODMAPs can further trigger GI symptoms in athletes with impaired GI function or in IBS patients [187]. In addition, foods high in FODMAPs can also cause upper-GI symptoms, such as stomach swelling due to the high consumption of fructose and glucose [184]. For example, upper-GI distress syndromes such as bloating, nausea, and stomach pain/cramps are common in cyclists, which can impair performance and well-being during exercise and daily life [188]. The potential efficiency of a low-FODMAP diet on exercise-induced GI symptoms has been studied in four studies, two randomized controlled crossover studies [67,70], and two case reports [68,69]. All studies suggested the low-FODMAP diet as an efficient treatment for reducing exercise-associated GI symptoms. A case study investigating a multisport athlete with exercise-induced GI symptoms showed that a short-term (6 day) restriction of foods high in FODMAPs (from 81.0 ± 5.0 g to 7.2 ± 5.7 g·day^−1^) resulted in a decrease in GI symptoms both during exercise and daily life of the athlete [69]. Another case report evaluated a long-term (4 week restriction of foods high in FODMAPs followed by reintroduction of foods high in FODMAPs for 6 weeks) low-FODMAP application before an aggressive multistage ultra-marathon race [68]. Apart from severe nausea, minimal GI symptoms including bloating and flatulence were observed throughout the race. Examining the influence of a 6-day low-FODMAP diet on recreationally competitive athletes with non-clinical GI symptoms in a single-blind, crossover design, Lis et al. [67] reported a significant decrease in exercise-induced GI symptoms, particularly in flatulence, urge to defecate, loose stool, and diarrhea, in nine of 11 athletes after the low-FODMAP trial. Another well-designed crossover study also applied 1 day low-FODMAP or high-FODMAP diet before exertional-heat stress to evaluate its impact on GI integrity, functions, and discomfort [70]. An exercise protocol that includes 2 h of work at 65% VO_2_max at 35 °C ambient temperature was applied after the diet applications. The study findings indicated that lower exercise-induced GI symptoms and I-FABP concentrations were observed after 1 day low-FODMAP diet, suggesting that 1 day low-FODMAP diet provided a crucial advantage by decreasing exercise-associated disruption of GI integrity, and attenuating GI symptoms [70]. Therefore, studies evaluating exertional-heat stress during long-term exercise have administered a 24 h low-FODMAP diet as a control diet to eliminate GI symptoms associated with food and fluid intake [189,190,191].

It should be noted that endurance athletes typically eat foods high in FODMAPs [8]. A study investigating the content of FODMAPs in various sports foods has shown that FODMAPs are often included in sports foods, such as dry dates (fructans), fructose, inulin (fructans), honey (fructose), and chicory root (oligosaccharides) [8]. Therefore, sports food alternatives low in FODMAPs could be a better choice for endurance athletes, in particular, those who have previously experienced GI symptoms.

A meta-analysis of nine randomized trials reported the administration of a low-FODMAP diet for short-term attenuated GI symptoms, abdominal pain, and quality of life in patients with IBS [192]. However, 25% of patients did not respond to the diet, and responders experienced the diarrhea-predominant type of IBS. These findings suggest that the higher response rate of diarrhea-type IBS may be due to osmotic changes in the gut following a low-FODMAP diet [192]. Note that runner’s diarrhea is known to be one of the most common exercise-related GI problems [168]; the chances of responding positively to a low-FODMAP diet are high in endurance athletes, especially athletes with runner’s diarrhea.

Exercise-induced GI symptoms may become detectable after intense exercise, affecting recovery and refueling periods [188]. The management of this process becomes crucial in multistage events that last multiple stages in a day or over multiple days [112,193]. In endurance athletes with exercise-induced GI symptoms, the FODMAPs restriction may also be needed for the post-exercise period [129], which is crucial to provide optimal nutrient delivery to the body after exercise, particularly intense training periods.

Taken together, as all studies on a low-FODMAP diet and exercise-associated GI symptoms confirmed the efficiency of the diet, we can consider a low-FODMAP diet as an efficient therapy to attenuate exercise-associated GI symptoms. However, the response rate to the low-FODMAP diet should also be determined before planning any long-term low-FODMAP diet application for endurance athletes.

#### 3.5.2. Potential Risks to Consider When Applying a Low-FODMAP Diet

A low-FODMAP diet may result in decreased consumption of prebiotics, which is highly recommended for maintaining a healthy gut microbiome [178]. Additionally, adherence to the diet may be problematic for athletes due to difficulties during the application process [3].

By assessing the low-FODMAP diet based on nutrients instead of general composition, we can realize that complex polysaccharides, the most significant prebiotic metabolites, are restricted with the diet, thus negatively affecting microbiome composition [194]. Although highlighted in several studies on humans with IBS [195,196,197], randomized controlled studies are needed to investigate the gut microbiome and low-FODMAP diet to evaluate the potential effects of a low-FODMAP diet in endurance athletes.

A low-FODMAP diet should not only attenuate GI problems but provide sports-specific nutrients and energy intake efficiently as well [171]. Subjects that fail to show any improvement during the first phase of long-term FODMAP application should not continue the diet [3]. Additionally, the reintroduction phase should be carefully applied to subjects by trained dietitians and professionals to identify which foods high in FODMAPs cause these symptoms, and personalize the diet to attenuate IBS symptoms, and thereby maintain healthy gut functions [8].

A general recommendation to reduce the FODMAP content of the diet consists of reducing FODMAP intake from 15–30 g FODMAP·day^−1^ to 5–18 g FODMAP·day^−1^ [198]. It is recommended for patients with IBS that less than 0.5 g FODMAP per meal or less than 3 g per day be consumed [199]. However, endurance athletes with exercise-induced GI symptoms consume 2-fold higher FODMAPs than the diet classified as high in FODMAPs in clinical research (up to 43 g·day^−1^) [67]. Therefore, foods high in FODMAPs could be a contributing factor for exercise-induced GI symptoms. A recent study on athletes reported that 55% (*n* = 910) of athletes removed at least one high FODMAP from their diet to attenuate exercise-induced GI symptoms, and approximately 85% reduced GI symptoms by removing food from their diet [171]. Lactose is often reported as the most problematic nutrient high in FODMAPs [163]. The most frequently eliminated foods are reported as lactose (86%), GOS (23.9%), fructose (23.0%), fructans (6.2%), and polyols (5.4%). Therefore, before strict FODMAP restriction, it should be considered that lactose and fructose are the most common inductors for GI distress [200]. Lactose consumption of athletes may be greater than that in the general population due to high protein ingredients, good sources of calcium, and rehydration [69]. Furthermore, higher fructose consumption may be greater in endurance athletes, especially during exercise due to sufficient energy supply during long-duration (> 90 min.) events or training [201]. Higher fructose intake may be more likely to trigger exercise-induced GI symptoms [202]. Therefore, just reducing or eliminating lactose and fructose instead of all high FODMAPs may inhibit the detrimental gut alterations and may solve the GI problems in endurance athletes.

## 4. Conclusions

This review discusses in detail the effectiveness of five popular diets, namely vegetarian diets, HFD, IF, GFD, and the low-FODMAP diet, on endurance performance and metabolism. Considering all findings from the review, all five diets discussed in detail appear to have both beneficial and detrimental effects on endurance performance (Figure 1). For vegetarian diets, we suggest that when adjusting the athlete’s diet a sports dietitian is to (a) determine which vegetarian diet the athlete is consuming; (b) control the athlete’s micronutrients and related biomarkers, especially vitamin B12, folate, vitamin D and iron; (c) regulate the athlete’s energy needs and all macro and micronutrient needs to prevent any deficiency, and (c) monitor the diet consumption and adjust it according to the needs based on individual- and sports-specific needs. While reviews of the HFD and sports performance have controversial results, the scientific evidence on the effectiveness of HFD on endurance performance is not strong enough to recommend these diets to endurance athletes. The evidence for IF diets and endurance performance and health-related parameters also needs to be improved by further investigation. We need more evidence before recommending the IF diet to endurance athletes. Considering all the relevant study results [66,68,69,70], we can say that a low-FODMAP diet may benefit more from GFD unless athletes have celiac disease. However, it should be kept in mind that the implementation steps of the low-FODMAP diet are complex and require careful monitoring by a trained dietitian. In addition, only lactose and fructose elimination from the diet should be considered in endurance athletes prior to adopting a low-FODMAP diet. We suggest that a short-term (1–6 days) low-FODMAP diet can be planned at first before planning a long-term strategy, especially before endurance racing or strenuous exercise. In summary, all five diets discussed in the review can be applied to endurance athletes in accordance with the athletes’ current metabolic demands. Before deciding on a popular diet, considering the current metabolic and sport-specific situation of endurance athletes will result in healthier and more beneficial results.

## Figures and Tables

**Figure 1 nutrients-13-00491-f001:**
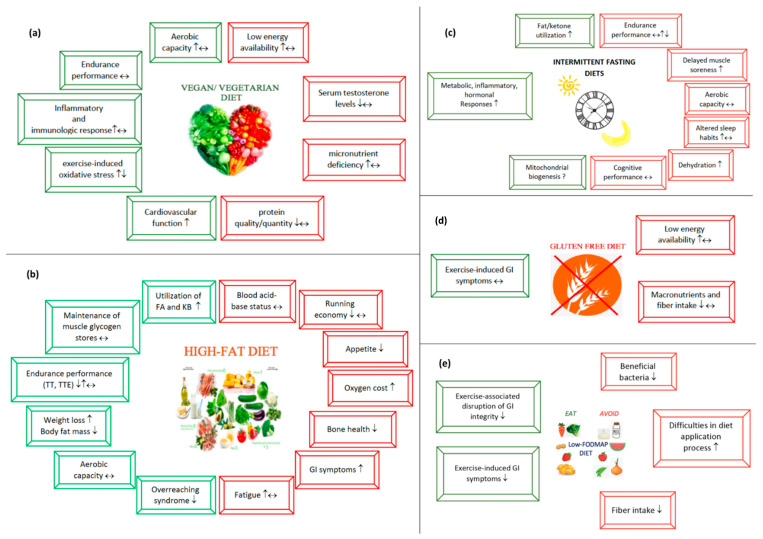
Possible beneficial and detrimental effects of popular diets on endurance athletes. Statements presented in green boxes show the beneficial effects of diets, while red boxes indicate the potential risks of diets. (**a**): Vegetarian diets; (**b**) high-fat diets; (**c**) Intermittent Fasting; (**d**) Gluten-free diet; (**e**) low-FODMAP diet. Abbreviations: URTI: Upper-respiratory tract infections; RED-S: relative energy deficiency syndrome; FA: fatty acids; KB: Ketone bodies; GI: Gastrointestinal; FODMAP: fermentable oligosaccharides, disaccharides, monosaccharides and polyols.

**Table 1 nutrients-13-00491-t001:** Studies investigating the potential effects of vegetarian, fasting, high-fat, gluten-free, and low-FODMAP diets on athletes’ endurance performance.

Subjects	Study Design	Diet/Application	Duration	Exercise Protocol(s)	Main Findings	Ref.
**High-Fat Diets**						
Endurance-trained male athletes (*n* = 20)	A non-randomized control trial	K-LCHF diet (*n* = 9; %CHO:fat:protein = 6:77:17) or HCD (*n* = 11; 65:20:14)	12 weeks	A 100-km TT performance, a 6-s sprint, and a CPT	↓ Body mass ↓ Body fat percentage ↑ Average relative power during the 6 s sprint sprint and CPT ↑ Fat oxidation during exercise ↔ 100 km TT endurance performance	[14]
Recreational male athletes (*n* = 14)	A randomized, crossover design	K-LCHF diet (<10%CHO, 75% fat) and 2 week HCD (>50% CHO), >2 weeks washout period in between	2 weeks	A 90-min bicycle ergometer exercise test at 60%Wmax	↓ Exercise-induced cortisol response; however, better results observed in HCD ↓ Exercise capacity ↑ Fat oxidation during exercise ↑ Perceived exertion after exercise ↔ Post-exercise s-IgA levels at week 2	[15]
Professional male race walkers (*n* = 25)	A mix of repeated-measures and parallel-group design	K-LCHF diet (*n* = 10; 75–80% FAT, <50 g CHO, 17% protein), HCD (*n* = 8; 60–65% CHO, 20% FAT, 15–20% protein), or PCD, (*n* = 7; 60–65% CHO, 20% FAT, 15–20% protein)	3 weeks	- A graded walking economy and VO_2_peak test on treadmill - A 10-km race walk (field) - A 25-km standardized race walk	↔ VO_2_peak ↓ 10 km race walk performance ↑ Perceived exertion after exercise ↑ Oxygen cost ↑ Fat oxidation during exercise	[16]
Male and female elite race walkers (*n* = 24)	A mix of repeated-measures and parallel-group design	K-LCHF diet (*n* = 9; 75–80% FAT, <50 g CHO, 15–20% protein), HCD (*n* = 8; 60–65% CHO, 20% FAT, 15–20% protein), or PCD, (*n* = 7; 60–65% CHO, 20% FAT, 15–20% protein)	3 weeks	- A graded walking economy and VO_2_peak test on treadmill	↔ VO_2_peak ↔ Blood acid-base status	[17]
Endurance-trained male athletes (*n* = 8)	A randomized repeated-measures crossover study	K-LCHF diet (75–80% FAT, <50 g CHO, 15–20% protein), HCD (43% CHO, 38% FAT, 19% protein)	4.5 weeks	- A graded metabolic test to exhaustion - A TTE performance at 70% VO_2_max	↔ TTE performance ↔ Perceived exertion after exercise ↓ Exercise efficiency above 70% VO_2_max ↔Exercise efficiency above 70% VO_2_max	[18]
Recreationally competitive male runners (*n* = 8)	A pre–post-test	K-LCHF diet (<50 g CHO, 70% FAT (ad libitum), or HCD (habitual diet defined as moderate to high CHO)	3 weeks	- Five 10-min running bouts at multiple individual race paces in the heat, 20-min rest, then a 5 km TT performance after 50 min of running in challenging environmental conditions	↔ 5 km TT performance ↔ Perceived exertion after exercise ↑ Fat oxidation during exercise ↓ Body mass ↓ Skinfold thickness ↔ Exercise-induced cardiorespiratory, thermoregulatory, or perceptual responses	[19]
Elite male cyclists (*n* = 5)	A pre–post-test	K-LCHF diet (<20 g CHO, 85% FAT, 15% protein) for 3 weeks immediately after a 1 week HCD (66% CHO, 33%FAT, 1.75 g protein/kg BW/d)	4 weeks (3 weeks LCKD after 1 week HCD)	- A VO_2_max test on cycle ergometry - A TTE test at 60–65% VO_2_max at two time points: after HCD and K-LCHF diet	↔ VO_2_max ↔ TTE performance ↑ Fat oxidation ↔ Blood glucose levels during TTE performance	[20]
Recreational athletes (*n* = 5)	Case study	K-LCHF diet (ad libitum FAT, <50 g CHO, 1.75 g protein/kg BW/d)	10 weeks	- A TTE performance - A peak power test - A VO_2_max test	↓ TTE performance ↑ Fat oxidation during exercise even at higher intensities ↓ Body mass ↓ Skinfold thickness	[21]
Endurance-trained male athletes (*n* = 8)	A randomized, repeated-measures, crossover study	K-LCHF diet (75–80% FAT, <50 g CHO, 15–20% protein), HCD (43% CHO, 38% FAT, 19% protein)	4.5 weeks	- A graded metabolic test to exhaustion - A TTE performance at 70% VO_2_max	Preservation of mucosal immunity ↑ Both pro- and anti-inflammatory T-cell-related cytokine responses to a multiantigen in vitro	[22]
Elite race walkers (*n* = 25)	A mix of repeated-measures and parallel-group design	K-LCHF diet (*n* = 10; 75–80% FAT, <50 g CHO, 17% protein), HCD (*n* = 8; 60–65% CHO, 2% FAT, 15–20% protein), or PCD, (*n* = 7; 60–65% CHO, 20% FAT, 15–20% protein)	3.5 weeks	- A graded walking economy and VO_2_peak test - A 10-km track race After CHO re-adaptation: - A 20-km race walking	↔ VO_2_peak ↓ 10 km race walk performance ↑ Perceived exertion after exercise ↑ Oxygen cost ↑ Whole-body fat oxidation	[23]
Male ultra-endurance runners (*n* = 20)	A cross-sectional study design	K-LCHF diet (*n* = 10, 10:19:70) diet or Habitual high-CHO (*n* = 10, %CHO:protein:fat = 59:14:25) diet	An average of 20 months (range 9–36 months)	- A graded exercise test - A 3-h run at 65% VO_2_max on a treadmill	↑ Fat oxidation ↔ Muscle glycogen utilization and repletion after 180 min of running and 120 min of recovery	[24]
Male competitive recreational distance runners (*n* = 7)	A randomized counterbalanced, crossover design	K-LCHF diet (*n* = 10; 75–80% FAT, <50 g CHO, 17% protein), or HCD (*n* = 8; 60–65% CHO, 20% FAT, 15–20% protein)	6 weeks	- A VO_2_max test - A 5-km TT performance (day 4, 14, 28, and 42)	↔ VO_2_max ↔ TT performance ↑ Fat oxidation	[25]
Endurance-trained male cyclists (*n* = 5)	Crossover design	A high-fat diet (70% FAT) or an equal-energy, high-carbohydrate diet (70% CHO)	2×2 weeks, 2 week washout period in between (ad libitum diet during washout period)	- Peak power output test - A cycling exercise to exhaustion at 90% VO_2_max, 20-min rest, and followed with a cycling exercise to exhaustion at 50% VO_2_max	↑ TTE performance during MIE ↔ Endurance performance during HIE ↑ Fat oxidation	[26]
Highly trained male ultra-endurance runners (*n* = 20)	A cross-sectional study design	Habitual low CHO (*n* = 10; <20% CHO, >60% FAT) or high CHO (*n* = 10; >55% CHO)	At least 6 months		↑ Circulating total cholesterol, LDL-C, and HDL-C concentrations ↑ Fewer small, dense LDL-C particles	[27]
Trained male off-road cyclists (*n* = 8)	A crossover design	A mixed diet (%CHO:fat:protein = 50:30:20) or a NK-LCHF diet (15:70:15)	4 weeks	A continuous exercise protocol on a cycling ergometer with varied intensity (90 min at 85% LT, then 15 min at 115% LT)	↑ VO_2_max ↓ Body mass ↓ Body fat percentage ↑ Fat oxidation ↓ Post-exercise muscle damage ↓ CK and LDH concentration at rest and during the 105 min exercise protocol in the NK-LCHF diet trial	[28]
Endurance trained cyclists (*n* = 16)	A randomized, controlled study design	A NK-LCHF diet (19:69:10) or a habitual diet (%CHO:fat:protein = 53:30:13)	15 days	a 2.5-h constant-load ride at 70% VO_2_peak followed by a simulated 40-km cycling TT while ingesting a 10% 14C-glucose + 3.44% MCT emulsion at a rate of 600 mL/h	↑ Fat oxidation ↔ TT performance	[29]
Trained male cyclists (*n* = 9)	A repeated-measures, randomized, crossover study	2 × 0.35 g/kg KE or placebo (30 min before and 60 min after exercise)	Acute ingestion	A 85-min steady state exercise at 73% VO_2_max, followed by a 7 kJ/kg TT (~ 30 min)	↑ Transient type-I T-cell immunity at the gen level	[30]
Endurance-trained male and female athletes (male/female, 9/3)	A single-blind, randomized and counterbalanced, crossover design	KE (330 mg/kg BW of βHB containing beverage, or bitter-flavored placebo drink before exercise	Acute ingestion	An incremental bicycle ergometer exercise test to exhaustion	↔ Blood pH and HCO_3_ levels ↔ TTE performance	[31]
Endurance-trained athletes (male/female:5/1)	A single-blind, random order controlled, crossover design	A 400 mL, low-dose β-HB KME 252 mg/kg BW, “low ketosis”; a high-dose βHB KME (752 mg/kg BW, “high ketosis”, or a bitter-flavored water (placebo)	Acute ingestion, 60 min prior to exercise	A 60-min continuous cycling exercise, consisting of 20 min intervals at 25%, 50% and 75% Wmax	↓ Contribution of exogenous βHB to overall energy expenditure ↑ Exercise efficiency when blood βHB levels above 2 mmol/L ↑ Nausea	[32]
High-performance athletes	Study 1: A randomized crossover design Study 2, 3 and 5: A randomized, single-blind, crossover design Study 4: A two-way crossover study	Study 1 (*n* = 6): A KE (573 mg/kg BW) drink at rest, and during 45 min of cycling exercise 40% and 75% of WMax; with 1 week washout period in between Study 2 (*n* = 10): - 96% of calories from CHO (dextrose = CHO), KE (573 mg/kg BW), or FAT before test Study 3 (*n* = 8): - 60% of calories from CHO and 40% of KE (573 mg/kg BW), a mixture of carbohydrates (CHO), or a no-calorie beverage with 1000 mg B3 before test Study 4 (*n* = 7): - 60% of calories from CHO (dextrose) and 40% from KE, or a mixture of CHOs, 50% of the drink consumed at baseline, the remaining 50% at 30 min, 60 min, and 90 min during exercise as equal aliquots Study 5 (*n* = 6 male, *n* = 2 female): - 60% of calories from CHO and KE (573 mg/kg BW), or a mixture of carbohydrates (CHO)	Acute ingestion	Study 1: - A 45-min cycling exercise at 40% and 75% WMax Study 2 and 3: - A fixed-intensity cycling exercise at 75% WMax for 60 min Study 4: - A fixed-intensity bicycle ergometry test at 70%VO_2_max for 2-h Study 5: - A 60-min steady state workload at 75% WMax followed by a blinded 30-min TT	↑ TT performance following 1 h of high-intensity exercise ↑ Fat oxidation ↓ Plasma lactate levels during exercise ↑ D-βHB oxidation according to exercise intensity (from 0.35 g/min at 40% WMax to 0.5 g/min at 75% WMax) ↔ Blood glucose levels	[33]
Trained male cyclists (*n* = 9)	A repeated-measures, randomized, crossover study	A drink containing 0.35 g/kg BW BD or placebo	Acute ingestion (30 min before and 60 min during 85 min of steady state exercise)	A steady state cycling at the power output eliciting 85% of their VT followed by a TT performance equivalent to 7 kJ/kg (~25–35 min)	↔ TT performance and average power output ↔ Blood glucose and lactate levels ↑ Fat oxidation ↑ GI symptoms	[34]
Elite male cyclists (*n* = 10)	A randomized crossover design	A 1,3-butanediol AcAc diester (2×250 mg/kg BW) or a viscosity and color-matched plasebo drink	Acute ingestion, ~30 min before and immediately prior to commencing the warm up	~A 31-km laboratory-based TT performance on a cycling ergometer	↓ TT performance ↑ GI symptoms (nausea and reflux) ↑ Fat oxidation	[35]
Male runners (*n* = 11)	A randomized crossover design	An energy matched ∼650 mL drink containing 60 g CHO + 0.5 g/kg BW 1.3-butanediol (CHO-BD) or 110 g ± 5 g CHO alone	Acute ingestion (50% after baseline measurements + 25% after 30 min of seated rest, + 25% after 10 min rest period after completing submaximal running)	A 60-min submaximal running, followed by a 5-km running time trial	↔ TT performance ↔ Overall lactate concentration ↑ Blood glucose levels after TT performance ↑ Fat oxidation	[36]
Highly trained male cyclists (*n* = 12)	A randomized crossover design	A KE drink (65 g (918,102 mg/kg, range: 722–1072 mg/kg) of KE [ 96% βHB] or a viscosity- and taste-matched placebo	Acute ingestion (at 60 and 20 min before and at 30 min during race)	A simulated cycling race, which consisted of a 3-h intermittent cycling, a 15-min time trial, and a maximal sprint	↔ High-intensity exercise performance in the final stage of the event ↑ Upper-abdominal discomfort ↓ Appetite after exercise ↔ Net muscle glycogen breakdown	[37]
Recreational male distance runners (*n* = 13)	A randomized, double-blind, placebo-controlled, cross- over design	Either one (KS1: 22.1 g) or two (KS2: 44.2 g) servings of the ketone supplement (βHB + MCT) or a flavor-matched placebo drink	Acute ingestion (60 min prior to exercise)	A 5-km running TT on a treadmill	↔ Post-exercise glucose concentration ↔ TT performance ↔ Perceived exertion after exercise Dose–response impact on cognitive function	[38]
Eight trained, middle- and long-distance runners (male/female, 7/1)	A double-blind, randomized crossover design	An 8% carbohydrate-electrolyte solution before and during exercise, either alone (CHO + PLA), or with 573 mg/kg of a ketone monoester supplement (CHO + KME)	Acute ingestion	A 60-min submaximal exercise at 65%VO_2_max immediately followed by a 10-km TT	↔ TT performance ↔ VO_2_max, running economy, RER, HR, perceived exertion ↔ Cognitive performance ↔ Plasma glucose and lactate levels ↑ Fat oxidation	[39]
Male and female elite race walkers	A non-randomized clinical trial	A K-LCHF diet (*n* = 18; 75–80% FAT, <50 g CHO, 15–20% PRO) followed by an acute CHO restoration, or HCD (*n* = 14; 60–65% CHO, 20% FAT, 15–20% PRO)	3.5 weeks	A hybrid laboratory/field test of 25 km (males) or 19 km (females) at around 50 km race pace at 75% VO_2_max	↓ Bone resorption markers at rest and post-exercise ↑ Bone formation markers at rest and throughout exercise Partial recovery of these effects following CHO restoration	[40]
Well-trained competitive male cyclists or triathletes (*n* = 7)	A randomized, crossover design	Day 1: a standard CHO diet (%CHO:fat:protein = 58:27:15) Day 2–7: either an HFD (16:69:15) or HCD (70:15:15) for 6 days Day 8: HCD (70:15:15)	6 day fat adaptation followed by 1 day CHO restoration, a 18 day washout period between	Day 9: A 4-h cycling ergometer at 65% VO_2_peak, followed by a 60-min TT	↔ TT performance ↑ Fat oxidation	[41]
Well-trained competitive male cyclists or triathletes (*n* = 8)	A randomized, crossover design	Day 1–5: either an HFD (%CHO:fat:protein = 19:68:13) or an HCD (74:13:13) Day 6: HCD (74:13:13)	5 day fat adaptation followed by 1 day CHO restoration, a 2 week washout period between	A 2-h cycling at 70% VO_2_max; followed by 7 kJ/kg TT	↔ TT performance ↑ Fat oxidation ↔ Muscle glycogen utilization ↔ Plasma glucose uptake	[42]
Well-trained competitive male cyclists or triathletes (*n* = 8)	A randomized, double-blind crossover design	Day 1–5: either an HFD (%CHO:fat:protein = 19:68:13) or an HCD (74:13:13) Day 6: HCD (74:13:13) Pre-exercise: a CHO breakfast (CHO 2 g/kg). During exercise: CHO intake (0.8 g/kg/h)	5 day fat adaptation followed by 1 day CHO restoration, a 2 week washout period between	A 2-h cycling at 70% VO_2_max; followed by 7 kJ/kg TT	↔ TT performance ↑ Fat oxidation	[43]
Well-trained competitive male cyclists or triathletes (*n* = 8)	A randomized, double-blind crossover design	Day 1–5: either an HFD (%CHO:fat:protein = 19:68:13) or an HCD (74:13:13) Day 6: HCD (74:13:13)	5 day fat adaptation followed by 1 day CHO restoration, a 2 week washout period between	A 60-min steady state ride at 70% VO_2_max	↓ Muscle glycogen utilization ↑ Fat oxidation ↑ Pre-exercise AMPK-1 and AMPK-2 activity ↓ Exercise-induced AMPK-1 and AMPK-2 activity	[44]
Endurance-trained male cyclists (*n* = 8)	A randomized, single-blind, crossover design	Day 1–6: either a NK- LCHF diet (%CHO:fat:protein = 16.8:68.2:15.0) or an HCD (67.8:17.1:15.1) Day 6: HCD (16.8:68.2:15.0)	6 day fat adaptation followed by 1 day CHO restoration, a 2 week washout period between	A 100-km TT on their bicycles; five 1 km sprint distances after 10, 32, 52, 72, and 99 km, four 4 km sprint distances after 20, 40, 60, and 80 km	↔ TT performance ↑ Fat oxidation ↓ 1 km sprint power ↔ Perceived exertion	[45]
Endurance-trained male cyclists (*n* = 5)	Randomized, crossover design	Either 10 day habitual diet (~30% fat), followed with 3 day HCD or 10 day high-fat diet (> 65% fat), followed by 3 day HCD 1 h prior to each trial: −400 mL 3.44% MCT (C_8–10_) solution During trial: 600 mL/h 10% glucose (^14^C) + 3.44% MCT solution	10 day HFD + 3 day HCD vs. 10 day habitual diet + 3 day HCD - Acute ingestion of MCT solution 1 h before trial and glucose + MCT solution during trial	A 150-min cycling at 70% VO_2_peak, followed immediately by a 20-km TT	↑ TT performance ↑ Fat oxidation ↓ Muscle glycogen utilization ↔ Body fat, BW	[46]
Endurance-trained male cyclists or triathletes (*n* = 7)	A randomized, double-blind crossover design	Day 1–5: either an HFD (%CHO:fat:protein = 19:68:13) or an HCD (74:13:13) Day 6: HCD (74:13:13)	5 day fat adaptation, a 2 week washout period between	A 20-min steady state cycling at 70% VO_2_peak, 1 min rest, a 1 min all-out sprint at 150% PPO, and followed by 4 kJ/kg TT	↑ Fat oxidation ↓ Glycogenolysis and PDH activation ↔ Muscle glycogen contents at rest	[47]
A lacto-ovo vegetarian athlete who adhered to an LCHF diet for 32 weeks	Case study	An LCHF diet for 32 weeks	32 weeks	Three professional races while on the LCHF diet in week 21, 24, and 32 (consumption of CHO before and during the race as advised)	↓ Half-ironman performance at week 21 ↓ Ironman performance at week 24 and 32 ↔ Exercise-induced GI symptoms	[48]
Trained male cyclists (*n* = 11)	A reference-controlled crossover (two treatment, two period), balanced, masked, single-center outpatient metabolic trial	HCD (% CHO:protein:fat = 73/14/12) for 2.5 days or HCD for first day and followed by the last 1.5 days with fat-enriched feeding (43/9/48)	2.5 days (1 day HCD, followed by lipid supplementation for 1.5 day), or 2.5 day HCD	Pre- and post-intervention; - A 3-h exercise on a bicycle ergometer at 50% Wmax post-intervention; - 20-km TT	↔ Perceived exertion after exercise ↔ Fat oxidation during prolonged exercise ↑ Replenishment of both glycogen content and IMCL stores ↔ TT performance	[49]
Trained male cyclists (*n* = 22)	A single-blind (clinical trial staff were blinded), 2-treatment crossover randomized clinical trial	An HCD, (CHO 7.4 g/kg BW, FAT 0.5 g/kg BW) for 2.5 days or a high-CHO fat-supplemented (HCF) diet ((first day similar with HCD, followed by 1.5 days with a replication of the HC diet with 240 g surplus fat (30% saturation)) distributed over the last 4 meals of the diet period	2.5 days (1 day HCD, followed by lipid supplementation for 1.5 day), or 2.5 day HCD	A fixed-task simulated TT lasting approximately 1-h A VO_2_peak test	↔ TT performance ↔ Fat oxidation during submaximal or 1 h TT exercise ↔ Reaction time throughout TT	[50]
Male collegiate long-distance athletes (*n* = 8)	A double-blind, placebo- controlled, crossover study design	3 days before the trial: an HCD (% CHO:fat:protein = 71:19:10) 4 h before exercise: HF meal (% CHO:fat:protein = 30:55:15) or HC meal (% CHO:fat:protein = 70:21:9) Immediately before exercise: - either maltodextrin jelly (M) or a placebo jelly (P) in the HF meal group - a P in the HC group	Acute ingestion (either HF meal or HC meal 4 h before exercise)	An 80-min fixed-load test on a treadmill at ~70 VO_2_max, followed with continuous endurance running to exhaustion at ~80% VO_2_max	↑ TTE performance in pre-exercise HF meal plus M consumption after CHO-loading ↑ Fat oxidation	[51]
**Vegetarian Diets**						
Vegan (*n* = 24), LOV (*n* = 26) and omnivorous (*n* = 26) recreational runners	A cross-sectional study design	Omnivorous, LOV or vegan diet for at least half a year	At least 6 months	An incremental exercise test on a bicycle ergometer	↔ maximum power output ↔ Exercise capacity ↔ Blood lactate and glucose concentration during incremental exercise	[52]
Vegan (*n* = 23), LOV (*n* = 25) and omnivorous (*n* = 25) recreational runners	A cross-sectional study design	Omnivorous, LOV or vegan diet for at least half a year	At least 6 months	An incremental exercise test on a bicycle ergometer	↑ exercise-induced MDA concentration in the vegan (+15% rise) and LOV (+24% rise) groups ↔ NO metabolism	[53]
Male endurance athletes (*n* = 8)	A crossover design	A mixed meat-rich diet (69% animal protein sources) or a LOV diet (82% vegetable protein sources)	2 ×6 weeks, 4 week washout period in between (ad libitum diet during washout period)	- A VO_2_max test	↔ Immunological parameters ↑ Fiber intake ↑ P/S ratio of fatty acids ↔ VO_2_max capacity	[54]
Omnivorous, lacto-ovo vegetarian, and vegan recreational runners (21–25 subjects, respectively)	A cross-sectional study	Omnivorous, lacto-ovo-vegetarian or vegan diet for at least half a year	At least 6 months	An incremental exercise test on a bicycle ergometer	↑ exercise-induced MDA concentration ↓ Sirtuin activities in vegans	[55]
A male vegan ultra-triathlete and a control group of 10 Ironman triathletes	Case report	A vegan ultra-triathlete adhered to a raw vegan diet and a control group of 10 Ironman triathletes adhered to a mixed diet	Vegan athlete living on a raw vegan diet for 6 years, vegan for 9 years and a vegetarian for 13 years	A Triple-Ironman distance (11.4 km swimming, 540 km cycling, and 126 km running)	↑ VO_2_max ↔ Exercise performance ↔ Exercise capacity ↔ Systolic and diastolic functions	[56]
A female vegan mountain biker	Case report	A vegan athlete living on a vegan diet for approximately 15 years	A vegan diet for approximately 15 years	The Transalp Challenge 2004 (altitude climbed, 22.500 m; total distance, 662 km, lasts approximately 8 days)	Successfully completing ultra-endurance mountain biking with a well-planned and implemented vegan diet	[57]
Vegetarian (*n* = 27) and omnivore (*n* = 43) elite endurance athletes	Cross-sectional study design	Vegetarian and omnivore endurance athletes who adhered to their respective diets for at least three months	At least three months	A VO_2_max test on the treadmill	↔ Exercise performance ↔ Protein intake (kg BW/day) ↑ VO_2_max (in females) ↔ VO_2_max (in males)	[58]
Vegan (*n* = 22) and omnivorous (*n* = 30) amateur runners	Cross-sectional study design	Vegan and omnivore athletes; diet adherence time not reported	-	VO_2_max and peak power output test on the treadmill	Better systolic and diastolic function ↑ VO_2_max	[59]
**Intermittent Fasting Diets**						
Well-trained, middle-distance runners (*n* = 18)	A non-randomized, controlled study	RIF vs. control	1 month	Beginning and at the end of Ramadan: - A VO_2_max test on the treadmill - A MVC testing - A 5-km TT	↓ TT exercise performance ↔ VO_2_max ↔ Running efficiency	[60]
Middle-distance athletes (*n* = 8)	Pre–post-test	RIF	1 month	5 days before, 7 and 21 days after Ramadan: - A maximal aerobic velocity test	↓ Nocturnal sleep time ↓ Energy intake ↔ BW and body fat percentage ↔ Testosterone/cortisol ratio ↑ Fatigue ↑ Transient alteration in circulating IL-6, adrenaline, noradrenaline levels	[61]
Elite under 23 cyclists (*n* = 16)	Parallel randomized trial	Time-restrictive eating (TRE) (16 h fasting, 8 h eating periods) or normal diet; both the same energy and macronutrient composition	4 weeks	Pre- and post-diet: - A VO_2_max test - A 45-min cycling ergometer at 45% peak power output	↔ VO_2_max ↔ endurance performance ↑ PPO/BW ratio ↓ BW and body fat percentage ↔ Fat-free mass	[62]
Male trained cyclists (*n* = 11)	A non-randomized repeated-measures experimental study design	Ramadan fasting (15 h 15 min fasting period)	29 days	A slow progressively increasing training load period (endurance training at first, and then intensity training included progressively)	↑ Perceived exertion ↑ DOMS ↔ Total sleep time ↓ duration of deep and REM sleep stages ↔ Cognitive performance	[63]
Adolescent male cyclists (*n* = 9)	A partially double-blind, placebo-controlled, randomized design	A CHO mouth rinse (with 25 mL of the solution) (CMR), a placebo mouth rinse (PMR), and a no-rinse (NOR) trial during Ramadan fasting state (fasting period ~13.5 h)	The last two weeks of Ramadan	A cycling exercise at 65% VO_2_peak for 30 min followed by a 10 km TT under hot (32 °C) humid (75%) condition	↑ TT performance in the CMR and PMR groups ↓ Perceived exertion in the CMR compared to the NOR ↔ Total sleep time	[64]
Trained male middle- and long-distance runners (*n* = 17)	A randomized, parallel-group, pre-and post-experimental design	A TRE (fasting: 16 h, ad libitum eating: 8 h) (*n* = 10) or normal diet (*n* = 7)	8 weeks	An incremental test until exhaustion	↓ BW ↔ VO_2_max ↔ Running economy ↔ Blood lactate, glucose, and insulin ↓ Daily energy intake	[65]
**Gluten-Free Diet**						
Non-coeliac or non-IBS competitive endurance cyclists (*n* = 13)	A controlled, randomized, double-blind, crossover study design	GFD or gluten-containing diet plus additional 2 gluten-free or gluten-containing food bars (total 16 g wheat gluten per day)	2 × 7 days, a 10 day washout period in between	A steady state cycling at 70% Wmax for 45 min followed by a 15 min TT	↔ TT performance ↔ GI symptoms ↔ Intestinal damage ↔ Well-being	[66]
**Low-FODMAP Diet**						
Recreationally competitive runners with non-clinical GI symptoms (5 males, 6 females)	A single-blind, crossover design	Either a high-FODMAP or a low-FODMAP (<0.5 g FODMAP/meal) diet	2×6 days, 1 day washout period in between	- A 5 × 1000-m run on day 4 - A 7-km threshold run on day 5	In the low-FODMAP group; ↔ Well-being ↓ GI symptoms	[67]
A female ultra- endurance runner	Case study	A 4 week low-FODMAP diet, (3.9 g FODMAP/day)	4 week low-FODMAP diet + 6 week reintroduction of high-FODMAP foods	A 6-day 186.7 km multistage ultra-marathon	Minimal GI symptoms ↑ Nausea ↓ Energy, protein, CHO, and water intake compared to the recommended guidelines	[68]
A recreationally competitive multisport athlete	Case study; a single-blind approach	A 6 day low-FODMAP diet (7.2 ± 5.7g FODMAPs/day) vs. habitual diet (81 ± 5 g FODMAPs/day)	6 days	Same training period both diet trial (Swim 60 min (day 1); cycle 60 min (day 2); rest (day 3); run intervals 70 min (day 4); cycle 180 min and steady state run 65 min (day 5) and; run intervals 65 min (day 6))	↓ Exercise-induced GI symptoms	[69]
Endurance runners (*n* = 18)	A double-blind randomized crossover design	A high- (46.9 ± 26.2 g FODMAP/day) or low- (2.0 ± 0.7 FODMAP/day) FODMAP diet	2 × 1 day; before each experimental trial	A 2-h running at 60% VO_2_max in 35 °C ambient temperature	In the low-FODMAP group; ↓ Exercise-associated disruption of GI integrity ↓ Exercise-associated GI symptoms ↓ Breath H_2_ concentration	[70]

↓: A significant decrease after the diet manipulation in the experimental group; ↑: A meaningful rise after the diet manipulation in the experimental group; ↔: No change after the diet manipulation in the experimental group. Abbreviations: K-LCHF: ketogenic low-carbohydrate, low-fat diet; NK-LCHF: non-ketogenic low-carbohydrate, low-fat diet CHO: carbohydrate; HCD: high-carbohydrate diet; TT: time trial; CPT: critical power test; s-IgA: serum immunoglobulin A; Wmax: maximal power output; VO_2_peak: peak oxygen uptake; VO_2_max: maximal oxygen uptake; PCD: periodized carbohydrate diet; TTE: time-to-exhaustion; MIE: moderate intensity exercise; HIE: high-intensity exercise; LDL-c: low-density lipoprotein; HDL-c: high-density lipoprotein; CK: creatine kinase; LDH: lactate dehydrogenase; SS: steady state; HCO_3_: hydrogen bicarbonate; KE: ketone ester; KME: ketone monoester; BW: body weight; βHB: (R)-3-hydroxybutyl (R)-3-hydroxybutyrate); VT: ventilatory threshold; GI: gastrointestinal; MCT: medium-chain triglycerides; RER: Respiratory exchange ratio; HR: heart rate; IMCL: Intra myocellular lipid; LOV: lacto-ovo-vegetarian; MDA: malondialdehyde; NO: nitric oxide; P/S ratio: polyunsaturated/saturated fatty acid ratio; MVC: Maximal Voluntary Isometric Contraction; IL-6: interleukine-6; PPO/BW ratio: peak power output/body weight ratio; DOMS: delayed onset muscle soreness; GFD: gluten-free diet; FODMAP: fermentable oligosaccharides, disaccharides, monosaccharides and polyols.

**Table 2 nutrients-13-00491-t002:** Types and application processes of new diets applied by endurance athletes.

Type	Other Terms Mentioned in Endurance Sport Research	Definition/Application	Ref.
**Vegetarian diets**			
Vegetarian diet	Vegetarian diet	Excludes all meats but may allow some animal products.	[99]
Ovo-vegetarian diet	Not detected	Excludes all meat and dairy products from the diet, but allows eggs.	[99]
Lacto-vegetarian diet	Not detected	Excludes all meat and eggs from the diet, but allows dairy products.	[99]
Lacto-ovo vegetarian diet	Lacto-ovo vegetarian diet	Excludes all types of meat from the diet, but allows the consumption of eggs and dairy products.	[99]
Pesco-vegetarian diet	Not detected	Excludes all animal products from the diet except fish.	[99]
Flexitarian diet	Not detected	A diet that flexible in terms of the consumption of animal products and allow to consume them occasionally.	[99]
**Vegan diet**			
Vegan diet	Vegan diet	Excludes all animal products from the diet.	[99]
**High-fat diets**			
Ketogenic low-CHO high-fat diet	Ketogenic diet; low-CHO ketogenic diet; ketogenic low-carbohydrate diet; keto-adaptation; high-fat diet; low-carbohydrate diet; low-carbohydrate, high-fat ketogenic diet	Consists of very low-CHO (20–50·g^−1^ day) and high-fat (75–80% of total energy) content with sufficient (15–20%) protein intake, resulting in increased ketone concentrations in blood named ketosis.	[5]
Non-ketogenic low-CHO high-fat diet	Non-ketogenic low-CHO high-fat diet, high-fat diet; low-carbohydrate diet	Consists of low-CHO (15–20% of total energy) and high-fat (60–65% of total energy) content with sufficient (15–20%) protein intake.	[5]
Acute ketone body supplementation	Ketone ester supplementation, ketone salt supplementation, a ketone monoester supplement, ketone diester ingestion, an exogenous ketone supplement	Creates exogenous ketosis, is applied in forms of either ketone salts or ketone esters.	[126]
CHO restoration following fat adaptation	Fat adaptation followed by CHO loading, keto-adaptation and glycogen restoration	A diet that is consumed a high-CHO diet for 1–3 days, and followed by a ketogenic or non-ketogenic high-fat diet for 5 to 14 days.	[5]
**Intermittent fasting diets**			
Complete alternate-day fasting	Intermittent fasting	Includes alternate fasting days (does not allow foods and drink consumption), and eating days (allow food and drink consumption ad libitum).	[127]
Modified fasting	Not detected	Includes a nocturnal fasting period of 16/18/20 h and an ad libitum-eating period of 8/6/4 h, (e.g., 5:2 diet, which includes 5 days (allows for food and drink consumption ad libitum) and 2 non-consecutive days (allows the consumption of 20–25% of energy needs ad libitum)).	[127]
Time-restricted eating	Time-restrictive eating (16/8)	Allows food or beverages at certain time periods, including regular, extended intervals (e.g., 16:8 diet with 16 h of fasting without energy intake and 8 h of food intake ad libitum).	[127]
Religious fasting	Ramadan intermittent fasting, Ramadan fast, Ramadan fasting	Comprises several fasting regimens based on specific religious and spiritual purposes (e.g., Ramadan fasting involving a fasting period from sunrise to sunset).	[127]
**Gluten-free diet**		Complete exclusion of gluten and gluten-containing products.	[128]
**Low-FODMAP diet**			
Long-term FODMAP elimination	A low-FODMAP diet, low-FODMAP foods	- (1) FODMAP restriction for 2 to 6 weeks from the athletes’ diet. - (2) reintroduction the restricted high-FODMAP foods step by step. - (3) individualize the athletes’ diet according to response against the first and second stages.	[129]
Short-term FODMAP elimination	24 h low-FODMAP diet	A strict FODMAP diet for 1 to 3 days before intensive training or races.	[129]

CHO: carbohydrate, FODMAP: fermentable oligosaccharides, disaccharides, monosaccharides and polyols.

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
