# Peer review of "Efficacy of Popular Diets Applied by Endurance Athletes on Sports Performance: Beneficial or Detrimental? A Narrative Review"

_nutrients, 2021, doi:10.3390/nu13020491_

Round 1
Reviewer 1 Report
The manuscript entitled “Efficacy of novel diets applied by endurance athletes on Sports Performance: Beneficial or detrimental? – A narrative review” presents interesting issue, but some areas must be corrected.
Major:
- Authors conducted a narrative review, that was based on some literature search. But the flaw of the presented manuscript is associated with the fact, that it presents a highly subjective review, not a systematic review. While the systematic review has a key role for broadening knowledge, the other reviews don’t have such role.
- Taking into account, that the Materials and methods section is not properly presented (it should be broaden), without necessary information, it is hard to understand which studies were included into review and why. Authors did not present properly key words, which were used during literature search, inclusion and exclusion criteria of references, information about the procedure of literature search conducted by them, number of chosen references, as well as information if some of them were excluded from the review and on the basis of which criteria. Authors can not indicate that they included some key words “and other terms” – all the actions should be properly described with all necessary details. As a number of recent publications that are related to the issue were not included, it is a serious problem.
- Authors presented some key words, but is seems that Authors did not search novel dietary approaches, but they assumed which are novel approaches to be included (e.g. ketogenic diet, high-fat diet, etc.) and searched only those terms. Such approach leads to preparing a highly subjective review (of only those approaches that were decided by Authors to be included).
- Authors do not present the current and comprehensive knowledge associated with the issue. It is associated with the fact that they did not include some important issues, while other were included even if they are not so crucial (e.g. Authors formulated their aim as associated with “novel” diets and afterwards included e.g. vegetarian diet – it is hard to conclude that vegetarian diet is something new)
- Authors are using a words and phrases which are catchy, but not justified, as they are not defined and should not be used in a scientific paper. One example is the term “plant-based diet” which is not defined in any way (see: https://www.ncbi.nlm.nih.gov/pmc/articles/PMC7533223/). Taking this into account, using this term (especially without a proper explanation) may lead to misunderstanding. Especially if Authors include not only vegetarian diets, but also flexitarian diets (as presented in Table 1), it may be the problem for readers.
- The information presented in the manuscript are not based on a properly chosen references – e.g. information that high-fat diets are “an effective dietary strategy for weight loss” requires proper reference of statement/ recommendation of using high-fat diets during body mass reduction, not a study on neurological diseases.
- Authors present improper information which are not based on any proper references – e.g. while presenting autophagy Authors indicate that “the popularity of fasting diets has increased following the Nobel Prize for the discovery of autophagy in 2016”. Autophagy was not discovered in 2016, as Ohsumi received Nobel Price while he discovered and elucidated mechanisms underlying autophagy (not discovered authophagy!) (see: https://www.nobelprize.org/prizes/medicine/2016/press-release/). Moreover – while presenting information about Nobel Prize in 2016, Authors refer 2 papers – published in 2004 and 1992 (how was it possible to anticipate Nobel Prize winner?)
- Including the reviews and meta-analysis into own review is also a highly controversial procedure – in many aspects, Authors just repeated the conclusions of other authors, without own analysis or conclusions.
- Very serious problem is associated with the fact that Authors did not present the existing recommendations which specify the possibility of using specific diets (e.g. https://pubmed.ncbi.nlm.nih.gov/19562864/).
- Last, but not least – such approach that was applied by Authors may have ethical problem. Authors formulate not justified conclusions. It may cause problems associated with scientific misunderstanding. Authors should imagine, that some people may read their review which is not justified, but readers may not know it. The presented information, while applied may cause a problem for a number of athletes.
Author Response
Response to Reviewer-1
The manuscript entitled “Efficacy of novel diets applied by endurance athletes on Sports Performance: Beneficial or detrimental? – A narrative review” presents interesting issue, but some areas must be corrected.
Dear reviewer,
Thank you for your valuable suggestions and comments. We noticed some gaps in our review and made a huge effort to improve it by rechecking the literature and adding more detail about the study objectives and, finally, providing a remarkable review to both researchers and athletes.
Answers to major revisions:
- To present Materials and Methods section properly, we add inclusion and exclusion criteria to this section. We also added all key words used for literature search as a Supplementary data.
- As working with over 150 endurance athletes, we also observed the applications of these diets. However, to define these diets, we searched the terms “diet” and “endurance athletes” in title, abstract, keyword in both PubMed and Cochrane databases. We reached 217 results in PubMed and 80 trials in Cochrane database. We define most recurrent diets in endurance athletes.
We changed the term “novel diet” to “popular diet” based on your comments.
- We reviewed the literature again, and discussed in the manuscript in detail.
- We included vegan and vegetarian diets into the study. Therefore, we changed the term “plant-based diet” to “vegan/vegetarian diets”.
We have included all the studies we have reached as a result of the literature review on endurance athletes and the aforementioned diets.
- We removed the data on Autophagy and Nobel Prize from the article.
- We add our perspectives based on the data presenting in the literature and also collected the study results in a Table and added as a Supplementary data (Table S1).
Your valuable contributions make great impact on us. We hope that we may able to answer all questions related to the review.
Kind regards
The authors
Reviewer 2 Report
28. overtraining has a negative meaning. Prefer prolonged training.
31-32. Please rephrase the sentence.
36. You must write that your search contains literature from training /races, as you did in “abstract”.
40. Give a range of the years that the studies used.
In “methodology”, report that your search targeted in races and training.
Also, are there any inclusion or exclusion criteria for the studies you used?
It would be useful to give the descriptive statistic of the number of studies that you used in every diet.
60. Give the definition of NURMI
60. Use “studied” not “investigated”
76. Do not use 1st plural.
124. Use the abbreviation. Please, check for the abbreviations in your text.
166-180. These sentences describe the beneficial effects of vegetarian and vegan dietary programs. In this section, I must read only the potential risks of plant-based diets.
205. per se? What means?
221. In this title, you use the full name’s parameter. In row 528 you use the abbreviation in the title. It is better to use the full name in titles and the abbreviations in the text.
263-264. Use the details of the diets out from parenthesis.
274-278. Give the full names of LCHF and HCLF. I cannot find them in the text.
284. Use fewer parenthesis in the text. Sometimes, long parenthesis makes it difficult to understand the meaning of the sentence.
306. Do you mean 2 mmol?
346-347. The beneficial role of HFD analyzed previously. It is not necessary to repeat it. Delete it.
440. Give the full name of PPAR-d.
557-558. Too long parenthesis.
In “Table d” you show that RED-S increases. In “3.4” chapter I cannot find anything about the increase of RED-S.
Generally, I would like to read some discussion of your results. I know that a Narrative’s review structure does not include a discussion section. Thus, I suggest you extend a little the end of every result section with the authors’ suggestions about the diets according to the literature. You can add an extra chapter for every diet, discussing your results.
Author Response
Response to Reviewer-2
Thank you for your valuable suggestions and comments that will permit to improve the paper quality.
Below are our responses to the reviewer’s comments.
- overtraining has a negative meaning. Prefer prolonged training.
Answer: As you suggested, we changed the term as “prolong training”.
31-32. Please rephrase the sentence.
Answer: We revised this sentences as follows: “…..At this point, it is of paramount importance that a popular diet should be scientifically proven before adopted in the athletic population…..”
You must write that your search contains literature from training /races, as you did in “abstract”.
Answer: As you suggested, we add a sentence in the abstract part as follows: “…..We reviewed scientific studies published from 2000 to 2021 that investigated the impact of these popular diets on endurance performance and health aspects of endurance athletes. ….”
Give a range of the years that the studies used.
Answer: We stated this information as follows”…. Apart from four important articles published before 2000 that we believe will add value to our review [9–12], we identified the studies published between 2000 and 2021. …”
In “methodology”, report that your search targeted in races and training.
Answer: We added the information to the methodology part as follows:”…… Using PubMed, Cochrane Library, and Web of Science databases, we aimed to identify studies on races and endurance training….”
Also, are there any inclusion or exclusion criteria for the studies you used?
It would be useful to give the descriptive statistic of the number of studies that you used in every diet.
Answer: As you suggested, we added our inclusion and exclusion criteria to the methodology section as follows: “…..We have included studies that are available in English, that clearly describe the applied diet, and investigate the effect of diet on endurance athletes as the primary goal. In addition, we included the studies in which diets were applied according to the dietary description indicated in Table 1. We excluded studies that did not explicitly address the impact of the diet on endurance performance or health-related parameters..……”
Give the definition of NURMI
Use “studied” not “investigated”
Answer: We used the verb as you suggested.
Do not use 1st plural.
Answer: We revised it as follows:”….. Given that the vegetarian diets are rich in carbohydrates (CHO) [18], such diets may offer more opportunities when considering races or training that can last at least six hours [2]……”
Use the abbreviation. Please, check for the abbreviations in your text.
Answer: We used abbreviation as you suggested.
166-180. These sentences describe the beneficial effects of vegetarian and vegan dietary programs. In this section, I must read only the potential risks of plant-based diets.
Answer: Thank you for your great comment. Adding this information to the risk section, we would like to highlight that a vegetarian diet could be a suitable option although it seems risky. We revised in line with our hypothesis “line 221-253”. However, if you would like us to exit this part, we will absolutely revise it again.
per se? What means?
Answer: We excluded the term as you suggested.
In this title, you use the full name’s parameter. In row 528 you use the abbreviation in the title. It is better to use the full name in titles and the abbreviations in the text.
Answer: We removed the abbreviation in the title and revised it as follows: “…..3.4.1.Why endurance athletes consider a gluten-free diet to be beneficial?....”
263-264. Use the details of the diets out from parenthesis.
Answer: We removed the parenthesis as you suggested. “….In addition to that, a recent study compared the efficiency of two energy-reduced (-500 kcal/day) diets, including a cyclical ketogenic reduction diet (CKD), defined as a high fat-low CHO (>30 g/day) diet for five days, followed by a high carb (8-10 g/body FFM) for two days, and a nutritionally balanced reduction diet (RD), a typical diet containing 55 % CHO, 15 % protein, and 30 % fat, on body composition and endurance performance in healthy young males …..”
274-278. Give the full names of LCHF and HCLF. I cannot find them in the text.
Answer: We gave the full names of K-LCHF and NK-LCHF as you suggested (line 308). Since HCLF was only mentioned here, we removed the abbreviation and mentioned it by its full name.
Use fewer parenthesis in the text. Sometimes, long parenthesis makes it difficult to understand the meaning of the sentence.
Answer: We rewrote this paragraph by adding extra information as one of reviewers suggested (line 422-429). We also used fewer parenthesis as you suggested.
Do you mean 2 mmol?
Answer: Thank you for your great attention. We defined SI Units of ketone bodies as mmol/L in the text.
346-347. The beneficial role of HFD analyzed previously. It is not necessary to repeat it. Delete it.
Answer: We deleted the sentence and started the paragraph with the following sentence: “….The deterioration of the running economy and increased oxygen cost during endurance exercise are considered to be major potential disadvantages of HFD….”
Give the full name of PPAR-d.
Answer: We added the full name of PPAR-d as you suggested.
557-558. Too long parenthesis.
Answer: We rewrote the sentence as follows: “….It is well-known that the "Belief effect" in athletes is an influential factor that can increase sports performance by 1 to 3%....”
In “Table d” you show that RED-S increases. In “3.4” chapter I cannot find anything about the increase of RED-S.
Answer: Thank you for your great attention. We removed the data from the figure.
Generally, I would like to read some discussion of your results. I know that a Narrative’s review structure does not include a discussion section. Thus, I suggest you extend a little the end of every result section with the authors’ suggestions about the diets according to the literature. You can add an extra chapter for every diet, discussing your results.
Answer: Thank you for your great contribution to our paper. With the advice of you and other reviewers, we discussed in depth after mentioning each potential benefit and risk.
Best regards
The authors
Reviewer 3 Report
Line 23 "We also discuss all the bright and dark aspects of these diets". Please change words "bright and dark aspects" to lights and shadows or other words more appropriated.
lines 54-56: remove names of famous athletes.
Lines 58-60. Describe percentage of only vegetarians and vegans athletes from the study of 422 runners . Pescatarian diet is not a vegetarian/vegan diet.
Lines 64-65 .
"3.1.1. The impact of plant-based diets on sports performance
Benefits of Plant-based diets"
Authors must be clear about whether they are going to describe plant based diets or vegetarian / vegan diets. They are not the same and if they want to include everything, they should change in line 51 title to "Plant based diets" instead of vegetarian/vegan diets or describe only vegetarian/vegan diets.
Table 1.
- Remove expression "has emerged in the last decades." from plant based diets.
- High-fat diets: if possible add similar information regarding nutrient contribution about all diets described in reference 32.
- drop a line in the second column that describes the diets to match the name of the diet in the first column
Lines 131-133. These data refer to a single study, reference 56, it is not generalized. Authors should put at the beginning of the sentence, "in a study"
Lines 137-143: The references included refer to articles of recommendations. Please add any study where it has been observed that the intakes of vegetarians are lower than the recommendations or studies with assessment of the nutritional status of athletes where the deficiency is verified. If they do not exist, the authors of this review should say so specifically at the beginning of line 133 "although intakes below the recommendations or nutritional deficiencies have not been observed" athletes may present a greater risk of ...etc.
Line 247: at the final of the sentence ..."on prolonged exercise" add "with anaerobic metabolism or high-intensity intermittent exercise".
Line 374 : The authors should explain in more detail some of the physiological consequences in the case of high protein intakes in HFD.
Line 449: In the study by Ba et al, the regulating factor of glycemia among endurance and sedentary athletes cannot be said to be the fasting by Ramadan because it is common to both but rather it should be exercise. Authors should clarify this point or remove sentence from manuscript.
Lines 466-499: The IF Ramadan diet is a specific type of IF diet where in many studies the dietary intake is ad libitum and not controlled. In this sense, I think that the authors should reduce or summarize the information described in articles with athletes in the Ramadan period. There is little information on IF diets in endurance, but many of the studies in Ramadan periods are not controlled trials.
Line 566: This statement extracted from reference 45 is questionable. There are many sources of carbohydrates gluten-free. There are no consistent studies showing energy deficiency in gluten-free diets. The authors should clarify that they "could lead to" in case of not well planned diets.
Line 593: word "bloating" is repeated.
Author Response
Response to Reviewer-3
Dear reviewer,
Thank you for your valuable suggestions and comments. We revised our paper as you suggested. All revisions are listed below:
Line 23 "We also discuss all the bright and dark aspects of these diets". Please change words "bright and dark aspects" to lights and shadows or other words more appropriated.
Answer: We changed the words "bright and dark aspects" to “lights and shadows” as you suggested.
lines 54-56: remove names of famous athletes.
Answer: We removed names of famous athletes as you suggested.
Lines 58-60. Describe percentage of only vegetarians and vegans athletes from the study of 422 runners. Pescatarian diet is not a vegetarian/vegan diet.
Answer: Thank you for your great attention. We cannot be able to define the exact percent of vegan, vegetarian, or pescatarian athletes due to the fact that the study did not categorize the information.
https://pubmed.ncbi.nlm.nih.gov/27077234/
Lines 64-65 .
"3.1.1. The impact of plant-based diets on sports performance
Benefits of Plant-based diets"
Authors must be clear about whether they are going to describe plant-based diets or vegetarian / vegan diets. They are not the same and if they want to include everything, they should change in line 51 title to "Plant based diets" instead of vegetarian/vegan diets or describe only vegetarian/vegan diets.
Answer: We aimed to describe vegetarian diets. Therefore, we chanced the term “Plant-based diets” to “Vegetarian diets” in the review as you suggested.
Table 1.
- Remove expression "has emerged in the last decades." from plant-based diets.
Answer: We removed the expression as you suggested.
- High-fat diets: if possible, add similar information regarding nutrient contribution about all diets described in reference 32.
Answer: Thank you for your attention. We replaced the citation with the correct citation we obtained the data from.
- drop a line in the second column that describes the diets to match the name of the diet in the first column
Answer: We added extra column to define the terms that describes the diets to match the name of the diet.
Lines 131-133. These data refer to a single study, reference 56, it is not generalized. Authors should put at the beginning of the sentence, "in a study"
Answer: We added the term “In a study” at the beginning of the sentence as you suggested.
Lines 137-143: The references included refer to articles of recommendations. Please add any study where it has been observed that the intakes of vegetarians are lower than the recommendations or studies with assessment of the nutritional status of athletes where the deficiency is verified. If they do not exist, the authors of this review should say so specifically at the beginning of line 133 "although intakes below the recommendations or nutritional deficiencies have not been observed" athletes may present a greater risk of ...etc.
Answer: Thank you for your contribution. We have added more details on studies evaluating the nutritional status of athletes (line 194-217).
Line 247: at the final of the sentence ..."on prolonged exercise" add "with anaerobic metabolism or high-intensity intermittent exercise".
Answer: Many thanks for your contribution. We added the term at the end of the sentence as you suggested (line 366-367).
Line 374 : The authors should explain in more detail some of the physiological consequences in the case of high protein intakes in HFD.
Answer: We added more detail on protein intake and its physiological impact on ketosis during KD as follows; “…..Another point regarding high protein intake during KD is that high protein consumption can disrupt ketosis by providing gluconeogenic precursors and thus inducing gluconeogenesis [135]. Therefore, moderate protein consumption is generally recommended during KDs. As we know that endurance athletes tend to consume more protein intake (1.2-2.0 g / kg BW / day) [136], this important effect of protein on ketosis should be kept in mind during the KD administration periods…..”
Line 449: In the study by Ba et al, the regulating factor of glycemia among endurance and sedentary athletes cannot be said to be the fasting by Ramadan because it is common to both but rather it should be exercise. Authors should clarify this point or remove sentence from manuscript.
Answer: We removed the sentence from the main text as you suggested.
Lines 466-499: The IF Ramadan diet is a specific type of IF diet where in many studies the dietary intake is ad libitum and not controlled. In this sense, I think that the authors should reduce or summarize the information described in articles with athletes in the Ramadan period. There is little information on IF diets in endurance, but many of the studies in Ramadan periods are not controlled trials.
Answer: Thank you for your great suggestion. We revised this section by adding the studies on endurance athletes and Ramadan IF at your suggestion and other reviewers.
Line 566: This statement extracted from reference 45 is questionable. There are many sources of carbohydrates gluten-free. There are no consistent studies showing energy deficiency in gluten-free diets. The authors should clarify that they "could lead to" in case of not well-planned diets.
Answer: Thank you for your contribution. We added the following explanation as follows: “…..Although a GFD limits the consumption of certain gluten-containing foods rich in CHO that could lead to an energy deficiency [166], there is insufficient data to investigate the effect of GFD on energy deficiency in endurance athletes. We recommend that more studies are required to be conducted on this topic, especially with a well-planned GFD for endurance athletes…..”
Line 593: word "bloating" is repeated.
Answer: We removed the repeated word “bloating”. Many thanks for the notice.
Round 2
Reviewer 1 Report
The manuscript entitled “Efficacy of popular diets applied by endurance athletes on sports performance: Beneficial or detrimental? – A narrative review” presents interesting issue, but some areas must be corrected.
Major:
- Authors conducted a narrative review, that was based on some literature search. But the serious flaw of the presented manuscript is associated with the fact, that it presents a highly subjective review, not a systematic review. While the systematic review has a key role for broadening knowledge, the other reviews don’t have such role.
- Authors have changes their title and indicated “popular diets” but they did not explain it in any way – what do Authors mean by “popular”? for whom? For which aspect?
- Taking into account, that the Materials and methods section is not properly presented (it should be broaden), without necessary information, it is hard to understand which studies were included into review and why. Authors did not present properly key words, which were used during literature search, inclusion and exclusion criteria of references, information about the procedure of literature search conducted by them, number of chosen references, as well as information if some of them were excluded from the review and on the basis of which criteria. In their previous version of the manuscript Authors indicated “and other terms” but now they were removed – it seems to be an ethical problem, as it seems that Authors used some terms and now they do not present them
- Authors presented some key words, but is seems that Authors did not search novel dietary approaches, but they assumed which are novel approaches to be included (e.g. ketogenic diet, high-fat diet, etc.) and searched only those terms. Such approach leads to preparing a highly subjective review (of only those approaches that were decided by Authors to be included).
- Including the reviews and meta-analysis into own review is also a highly controversial procedure – in many aspects, Authors just repeated the conclusions of other authors, without own analysis or conclusions.
- Last, but not least – such approach that was applied by Authors is a very serious ethical problem. Authors formulate not justified conclusions. It may cause not only problems associated with scientific misunderstanding, but it may also be life-threatening situation. Authors should imagine, that some people may read their review which is not justified, but readers may not know it. The presented information, while applied may cause a problem for a number of athletes.
Author Response
Response to reviewer 1
Dear Reviewer 1,
In this narrative review, we aimed to make interpretations by discussing the literature data, including research articles and case reports on endurance athletes, on diets in detail.
We add additional information about the selection processing of these diets in the material and method section as follows: “….We searched the terms “diet*”, "track-and-field", "runner*", "marathoner*", "cyclist", "cycling", "triathlete", "endurance", and “endurance athletes” in the title, abstract, keyword in both PubMed and Cochrane databases to define the applied diets between 2015 and 2021 in endurance athletes. We reached 217 results in PubMed and 80 trials in the Cochrane database. We defined most recurrent diets in endurance athletes, including "High CHO availability", "High-carbohydrate diet", "Ketogenic diet", "Low-CHO diet", "Low-CHO, high-fat diet", "Ketogenic low carbohydrate, high fat diet", "Low-carbohydrate ketogenic diet", "Low-carbohydrate, high fat, ketogenic diet", "High-fat, low carbohydrate diet", "Ketone ester supplementation", "time-restrictive eating", "Ketone supplementation", "Intermittent fasting", " fasting during Ramadan", "Vegan diet", "Lacto-Ovo vegetarian diet", "Vegetarian diet", "Low fermentable oligo-, di-, monosaccharide, and polyol diet", and "Gluten-free diet". Since all we know that high-carbohydrate diet is already well-proven to enhance endurance performance, we targeted to investigate other diets in-depth by categorizing them as "vegan/vegetarian diets", "high-fat diets", "intermittent fasting", "low-FODMAP diet, and "gluten-free diet"…..”
To be better defined, we can refer to these diets as "common" diets.
We define all key words into the Material and methods section. We defined inclusion and exclusion criteria of references, number of references.
We removed other terms due to the fact that we added all other terms as Supplementary data. Now, we added all terms into the Material and Methods section.
We added number of references into the Material and Methods section.
We reviewed all these diets with their beneficial and detrimental sides and suggested for future studies that all of these diets need further investigation. We indicated that any of these diets were not a certain solution for improving endurance performance and clearly defined that every endurance athlete should be careful when applying these diets.
Thank you for all your contributions for improving this review.
Kind regards
The authors
This manuscript is a resubmission of an earlier submission. The following is a list of the peer review reports and author responses from that submission.
Round 1
Reviewer 1 Report
This is an extensive review of the effects of diets, foods, nutrients, and xenobiotics on human athletic performance and fecal microbiota. While there are extensive citations and the literature is properly referenced, this reads like a series of blog posts in which several areas overlap and citations are made with commentary, though the authors don't seem to understand what they are stating or proposing (all sections describing cellular signalling mechanisms, the section on lipopolysaccharide, oxylipins, etc.)
The authors should consider reorganizing their content from the molecular level out to dietary patterns to improve objectivity and flow. Thus, all micronutrients would be discussed as well as polyphenols and other compounds found in supplements such as exogenous ketones. Next, macronutrients and fiber types can be discussed, followed my macronutrient distributions and finally dietary patterns.
Reviewer 2 Report
None.